# Solving Minimum-Cost Reach Avoid using Reinforcement Learning

**Oswin So***
Department of Aeronautics and Astronautics
MIT
oswinso@mit.edu

**Cheng Ge***
Department of Aeronautics and Astronautics
MIT
gec_mike@mit.edu

**Chuchu Fan**
Department of Aeronautics and Astronautics
MIT
chuchu@mit.edu

## Abstract

Current reinforcement-learning methods are unable to *directly* learn policies that solve the minimum cost reach-avoid problem to minimize cumulative costs subject to the constraints of reaching the goal and avoiding unsafe states, as the structure of this new optimization problem is incompatible with current methods. Instead, a *surrogate* problem is solved where all objectives are combined with a weighted sum. However, this surrogate objective results in suboptimal policies that do not directly minimize the cumulative cost. In this work, we propose **RC-PPO**, a reinforcement-learning-based method for solving the minimum-cost reach-avoid problem by using connections to Hamilton-Jacobi reachability. Empirical results demonstrate that RC-PPO learns policies with comparable goal-reaching rates to while achieving up to $57\%$ lower cumulative costs compared to existing methods on a suite of minimum-cost reach-avoid benchmarks on the Mujoco simulator. The project page can be found at https://oswinso.xyz/rcppo/.

## 1 Introduction

Many real-world tasks can be framed as a constrained optimization problem where reaching a goal at the terminal state and ensuring safety (i.e., reach-avoid) is desired while minimizing some cumulative cost as an objective function, which we term the *minimum-cost* reach-avoid problem.

The cumulative cost, which differentiates this from the traditional reach-avoid problem, can be used to model desirable aspects of a task such as minimizing energy consumption, maximizing smoothness, or any other pseudo-energy function, and allows for choosing the most desirable policy among many policies that can satisfy the reach-avoid requirements. For example, energy-efficient autonomous driving [1, 2] can be seen as a task where the vehicle must reach a destination, follow traffic rules, and minimize fuel consumption. Minimizing fuel use is also a major concern for low-thrust or energy-limited systems such as spacecraft [3] and quadrotors [4]. Quadrotors often have to choose limited battery life to meet the payload capacity. Hence, minimizing their energy consumption, which can be done by taking advantage of wind patterns, is crucial for keeping them airborne to complete more tasks. Other use-cases important for climate change include plasma fusion (reach a desired current, minimize the total risk of plasma disruption) [5] and voltage control (reach a desired voltage level, minimize the load shedding amount) [6].

---

*These authors contributed equally to this work

38th Conference on Neural Information Processing Systems (NeurIPS 2024).

If only a single control trajectory is desired, this class of problems can be solved using numerical trajectory optimization by either optimizing the timestep between knot points [7] or a bilevel optimization approach that adjusts the number of knot points in an outer loop [8, 9, 10]. However, in this setting, the dynamics are assumed to be known, and only a single trajectory is obtained. Therefore, the computation will needs to be repeated when started from a different initial state. The computational complexity of trajectory optimization prevents it from being used in real time. Moreover, the use of nonlinear numerical optimization may result in poor solutions that lie in suboptimal local minim [11].

Alternatively, to obtain a control policy, reinforcement learning (RL) can be used. However, existing methods are unable to directly solve the minimum-cost reach-avoid problem. Although RL has been used to solve many tasks where reaching a goal is desired, goal-reaching is encouraged as a *reward* instead of as a *constraint* via the use of either a sparse reward at the goal [12, 13, 14], or a surrogate dense reward [14, 15][1]. However, posing the reach constraint as a reward then makes it difficult to optimize for the cumulative cost at the same time. In many cases, this is done via a weighted sum of the two terms [18, 19, 20]. However, the optimal policy of this new *surrogate* objective may not necessarily be the optimal policy of the original problem. Another method of handling this is to treat the cumulative cost as a constraint and solve for a policy that maximizes the reward while keeping the cumulative cost under some fixed threshold, resulting in a new constrained optimization problem that can be solved as a constrained Markov decision process (CMDP) [21]. However, the choice of this fixed threshold becomes key: too small and the problem is not feasible, destabilizing the training process. Too large, and the resulting policy will simply ignore the cumulative cost.

To tackle this issue, we propose **Reach Constrained Proximal Policy Optimization(RC-PPO)**, a new algorithm that targets the minimum-cost reach-avoid problem. We first convert the reach-avoid problem to a reach problem on an augmented system and use the corresponding *reach* value function to compute the optimal policy. Next, we use a novel two-step PPO-based RL-based framework to learn this value function and the corresponding optimal policy. The first step uses a PPO-inspired algorithm to solve for the optimal value function and policy, conditioned on the cost upper bound. The second step fine-tunes the value function and solves for the least upper bound on the cumulative cost to obtain the final optimal policy. Our main contributions are summarized below:

- We prove that the minimum-cost reach-avoid problem can be solved by defining a set of augmented dynamics and a simplified constrained optimization problem.

- We propose RC-PPO, a novel algorithm based on PPO that targets the minimum-cost reach-avoid problem, and prove that our algorithm converges to a locally optimal policy.

- Simulation experiments show that RC-PPO achieves reach rates comparable with the baseline method with the highest reach rate while achieving significantly lower cumulative costs.

## 2 Related Works

**Terminal-horizon state-constrained optimization** Terminal state constraints are quite common in the dynamic optimization literature. For the finite-horizon case, for example, one method of guaranteeing the stability of model predictive control (MPC) is with the use of a terminal state constraint [22]. Since MPC is implemented as a discrete-time finite-horizon numerical optimization problem, the terminal state constraints can be easily implemented in an optimization program as a normal state constraint. The case of a flexible-horizon constrained optimization is not as common but can still be found. For example, one method of time-optimal control is to treat the integration timestep as a control variable while imposing state constraints on the initial and final knot points [7]. Another method is to consider a bilevel optimization problem, where the number of knot points is optimized for in the outer loop [8, 9, 10].

**Goal-conditioned Reinforcement Learning** There have been many works on goal-conditioned reinforcement learning. These works mainly focus on the challenges of tackling sparse rewards [12, 14, 23, 15] or even learning without rewards completely, either via representation learning objectives [24, 25, 26, 27, 28, 29, 30, 31, 32, 33] or by using contrastive learning to learn reward functions [34, 35, 36, 37, 38, 39, 40, 41, 42, 43], often in imitation learning settings [44, 45]. However,

---

[1]If the dense reward is not specified correctly, however, it can lead to unwanted local minima [14] that optimize the reward function in an undesirable manner, i.e., *reward hacking* [16, 17]

the *manner* in which these goals are reached is not considered, and it is difficult to extend these works to additionally minimize some cumulative cost.

**Constrained Reinforcement Learning**   One way of using existing techniques to approximately tackle the minimum-cost reach-avoid problem is to flip the role of the cumulative-cost objective and the goal-reaching constraint by treating the goal-reaching constraint as an objective via a (sparse or dense) reward and the cumulative-cost objective as a constraint with a cost threshold, turning the problem into a CMDP [21]. In recent year, there has been significant interest in deep RL methods for solving CMDPs [46, 47, 48]. While these methods are effective at solving the transformed CMDP problem, the optimal policy to the CMDP may not be the optimal policy to the original minimum-cost reach-constrained problem, depending on the choice of the cost constraint.

**State Augmentation in Constrained Reinforcement Learning**   To improve reward structures in constrained reinforcement learning, especially in safety-critical systems, one effective approach is state augmentation. This technique integrates constraints, such as safety or energy costs, into the augmented state representation, allowing for more effective constraint management through the reward mechanism [49, 50, 51]. While these methods enhance the reward structure for solving the transformed CMDP problems, they still face the inherent limitation of the CMDP framework: the optimal policy for the transformed CMDP may not always correspond to the optimal solution for the original problem.

**Reachability Analysis**   Reachability analysis looks for solutions to the reach-avoid problem. That is, to solve for the set of initial conditions and an appropriate control policy to drive a system to a desired goal set while avoiding undesireable states. Hamilton-Jacobi (HJ) reachability analysis [52, 53, 54, 55, 56] provides a methodology for the case of dynamics in *continuous-time* via the solution of a partial differential equation (PDE) and is conventionally solved via numerical PDE techniques that use state-space discretization [54]. This has been extended recently to the case of discrete-time dynamics and solved using off-policy [57, 58] and on-policy [59, 60] reinforcement learning. While reachability analysis concerns itself with the reach-avoid problem, we are instead interested in solutions to the *minimum-cost* reach-avoid problem.

## 3   Problem Formulation

In this paper, we consider a class of *minimum-cost* reach-avoid problems defined by the tuple $\mathcal{M} := \langle \mathcal{X}, \mathcal{U}, f, c, g, h \rangle$. Here, $\mathcal{X} \subseteq \mathbb{R}^n$ is the state space and $\mathcal{U} \subseteq \mathbb{R}^m$ is the action space. The system states $x_t \in \mathcal{X}$ evolves under the *deterministic* discrete dynamics $f : \mathcal{X} \times \mathcal{U} \to \mathcal{X}$ as

$$x_{t+1} = f(x_t, u_t). \tag{1}$$

The control objective for the system states $x_t$ is to reach the goal region $\mathcal{G}$ and avoid the unsafe set $\mathcal{F}$ while minimizing the cumulative cost $\sum_{t=0}^{T-1} c(x_t, \pi(x_t))$ under control input $u_t = \pi(x_t)$ for a designed control policy $\pi : \mathcal{X} \to \mathcal{U}$. Here, $T$ denotes the first timestep that the agent reaches the goal $\mathcal{G}$. The sets $\mathcal{G}$ and $\mathcal{F}$ are given as the 0-sublevel and strict 0-superlevel sets $g : \mathcal{X} \to \mathbb{R}$ and $h : \mathcal{X} \to \mathbb{R}$ respectively, i.e.,

$$\mathcal{G} := \{x \in \mathcal{X} \mid g(x) \le 0\}, \quad \mathcal{F} := \{x \in \mathcal{X} \mid h(x) > 0\} \tag{2}$$

This can be formulated formally as finding a policy $\pi$ that solves the following constrained *flexible* final-time optimization problem for a given initial state $x_0$:

$$\min_{\pi, T} \quad \sum_{t=0}^{T-1} c\big(x_t, \pi(x_t)\big) \tag{3a}$$

$$\text{s.t.} \quad x_T \in \mathcal{G}, \tag{3b}$$

$$x_t \notin \mathcal{F} \quad \forall t \in \{0, \dots, T\}, \tag{3c}$$

$$x_{t+1} = f\big(x_t, \pi(x_t)\big). \tag{3d}$$

Note that as opposed to either traditional finite-horizon constrained optimization problems where $T$ is fixed or infinite-horizon problems where $T = \infty$, the time horizon $T$ is also a decision variable. Moreover, the goal constraint (3b) is only enforced at the terminal timestep $T$. These two differences prevent the straightforward application of existing RL methods to solve (3).

## 3.1 Reachability Analysis for Reach-Avoid Problems

In discrete time, the set of initial states that can reach the goal $\mathcal{G}$ without entering the avoid set $\mathcal{F}$ can be represented by the 0-sublevel set of a reach-avoid value function $V_{g,h}^\pi$ [58]. Given functions $g, h$ describing $\mathcal{G}$ and $\mathcal{F}$ and a policy $\pi$, the reach-avoid value function $V_{g,h}^\pi : \mathcal{X} \to \mathbb{R}$ is defined as

$$V_{g,h}^\pi(x_0) = \min_{T \in \mathbb{N}} \max \left\{ g(x_T^\pi), \ \max_{t \in \{0,\dots,T\}} h(x_t^\pi) \right\}, \tag{4}$$

where $x_t^\pi$ denote the system state at time $t$ under a policy $\pi$ starting from an initial state $x_0^\pi = x_0$. In the rest of the paper, we suppress the argument $x_0$ for brevity whenever clear from the context. It can be shown that the reach-avoid value function satisfies the following recursive relationship via the reach-avoid Bellman equation (RABE) [58]

$$V_{g,h}^\pi(x_t^\pi) = \max \left\{ h(x_t^\pi), \ \min\{g(x_t^\pi), V_{g,h}^\pi(x_{t+1}^\pi)\} \right\}, \quad \forall t \geq 0. \tag{5}$$

The Bellman equation (5) can then be used in a reinforcement learning framework (e.g., via a modification of soft actor-critic[61, 62]) as done in [58] to solve the reach-avoid problem.

Note that existing methods of solving reach-avoid problems through this formulation focus on minimizing the value function $V_{g,h}^\pi$. This is not necessary as any policy that results in $V_{g,h}^\pi \leq 0$ solves the reach-avoid problem, albeit without any cost considerations. However, it is often the case that we wish to minimize a cumulative cost (e.g., (3a)) on top of the reach-avoid constraints (3b)-(3c) for a *minimum-cost* reach-avoid problem. To address this class of problems, we next present a modification to the reach-avoid framework that additionally enables the minimization of the cumulative cost.

## 3.2 Reachability Analysis for Minimum-cost Reach-Avoid Problems

We now provide a new framework to solve the minimum-cost reach-avoid by lifting the original system to a higher dimensional space and designing a set of augmented dynamics that allow us to convert the original problem into a reachability problem on the augmented system.

Let $\mathbb{I}$ denote the shifted indicator function defined as

$$\mathbb{I}_{b \in B} := \begin{cases} +1 & b \in B, \\ -1 & b \notin B. \end{cases} \tag{6}$$

Define the *augmented* state as $\hat{x} = (x, y, z) \subseteq \hat{\mathcal{X}} := \mathcal{X} \times \{-1, 1\} \times \mathbb{R}$. We now define a corresponding augmented dynamics function $f' : \hat{\mathcal{X}} \times \mathcal{U} \to \hat{\mathcal{X}}$ as

$$\hat{f}(x_t, y_t, z_t, u_t) = \big(f(x_t), \ \max\{\mathbb{I}_{f(x_t) \in \mathcal{F}}, \ y_t\}, \ z_t - c(x_t, \ u_t)\big), \tag{7}$$

where $y_0 = \mathbb{I}_{x_0 \in \mathcal{F}}$. Note that $y_t = 1$ if the state has entered the avoid set $\mathcal{F}$ at some timestep from 0 to $t$ and is *unsafe*, and $y_t = 0$ if the state has not entered the avoid set $\mathcal{F}$ at any timestep from 0 to $t$ and is *safe*. Moreover, $z_t$ is equal to $z_0$ minus the cost-to-come, i.e., for state trajectory $x_{0:t}$ and action trajectory $u_{0:t}$, i.e.,

$$z_{t+1} = z_0 - \sum_{k=0}^{t} c(x_t, u_t). \tag{8}$$

Under the augmented dynamics, we now define the following augmented goal function $\hat{g} : \hat{\mathcal{X}} \to \mathbb{R}$ as

$$\hat{g}(x, y, z) := \max\{g(x), \ Cy, \ -z\}, \tag{9}$$

where $C > 0$ is an arbitrary constant.[2] With this definition of $\hat{g}$, an augmented goal region $\hat{\mathcal{G}}$ can be defined as

$$\hat{\mathcal{G}} := \{\hat{x} \mid \hat{g}(\hat{x}) \leq 0\} = \{(x, y, z) \mid x \in \mathcal{G}, \ y = -1, \ z \geq 0\}. \tag{10}$$

In other words, starting from initial condition $\hat{x}_0 = (x_0, y_0, z_0)$, reaching the goal on the augmented system $\hat{x}_T \in \hat{g}$ at timestep $T$ implies that 1) the goal is reached at $x_T$ for the original system, 2) the state trajectory remains safe and does not enter the avoid set $\mathcal{F}$, and 3) $z_0$ is an upper-bound on the total cost-to-come: $\sum_{t=0}^{T-1} c(x_t, u_t) \leq z_0$. We call this the upper-bound property. The above intuition on the newly defined augmented system is formalized in the following theorem, whose proof is provided in Appendix D.1.

---

[2]In practice, we use $C = \max_{x \in \mathcal{X}} g(x)$.

**Theorem 1.** For given initial conditions $x_0 \in \mathcal{X}$, $z_0 \in \mathbb{R}$ and control policy $\pi$, consider the trajectory for the original system $\{x_0, \ldots x_T\}$ and its corresponding trajectory for the augmented system $\{(x_0, y_0, z_0), \ldots (x_T, y_T, z_T)\}$ for some $T > 0$. Then, the reach constraint $x_T \in \mathcal{G}$ (3b), avoid constraint $x_t \notin \mathcal{F} \; \forall t \in \{0, 1, \ldots, T\}$ (3c) and the upper-bound property $z_0 \geq \sum_{k=0}^{T-1} c(x_k, \pi(x_k))$ hold if and only if the augmented state reaches the augmented goal at time $T$, i.e., $(x_T, y_T, z_T) \in \hat{\mathcal{G}}$.

With this construction, we have folded the avoid constraints $x_t \notin \mathcal{F}$ (3c) into the reach specification on the augmented system. In other words, solving the reach problem on the augmented system results in a reach-avoid solution of the original system. As a result, we can simplify the value function (4) and Bellman equation (5), resulting in the following definition of the reach value function $\tilde{V}_{\hat{g}} : \hat{\mathcal{X}} \to \mathbb{R}$

$$\tilde{V}_{\hat{g}}^{\pi}(\hat{x}_0) = \min_{t \in \mathbb{N}} \hat{g}(\hat{x}_t^{\pi}). \tag{11}$$

Similar to (4), the 0-sublevel set of $\tilde{V}_{\hat{g}}$ describes the set of augmented states $\hat{x}$ that can reach the augmented goal $\hat{\mathcal{G}}$. We can also similarly obtain a recursive definition of the reach value function $\tilde{V}_{\hat{g}}$ given by the reachability Bellman equation (RBE)

$$\tilde{V}_{\hat{g}}^{\pi}(x_t^{\pi}, y_t^{\pi}, z_t^{\pi}) = \min\left\{ \hat{g}(x_t^{\pi}, \ y_t^{\pi}, z_t^{\pi}), \tilde{V}_{\hat{g}}^{\pi}(x_{t+1}^{\pi}, y_{t+1}^{\pi}, z_{t+1}^{\pi}) \right\} \quad \forall t \geq 0, \tag{12}$$

whose proof we provide in Appendix D.2.

We now solve the minimum-cost reach-avoid problem using this augmented system. By Theorem 1, the $z_0$ is an upper bound on the cumulative cost to reach the goal while avoiding the unsafe set if and only if the augmented state $\hat{x}$ reaches the augmented goal. Since this upper bound is tight, the least upper bound $z_0$ that still reaches the augmented goal thus corresponds to the minimum-cost policy that satisfies the reach-avoid constraints. In other words, the minimum-cost reach-avoid problem for a given initial state $x_0$ can be reformulated as the following optimization problem.

$$\min_{\pi, \, z_0} \quad z_0 \tag{13a}$$

$$\text{s.t.} \quad \tilde{V}_{\hat{g}}^{\pi}(x_0, \mathbb{I}_{x_0 \in \mathcal{F}}, z_0) \leq 0. \tag{13b}$$

We refer to Appendix B for a detailed derivation of the equivalence between the transformed Problem 13 and the original minimum-cost reach-avoid Problem 3.

*Remark* 1 (Connections to the epigraph form in constrained optimization). The resulting optimization problem (13) can be interpreted as an epigraph reformulation [63] of the minimum-cost reach-avoid problem (3). The epigraph reformulation results in a problem with *linear* objective but yields the same solution as the original problem [63]. The construction we propose in this work can be seen as a *dynamic* version of this epigraph reformulation technique originally developed for static problems and is similar to recent results that also make use of the epigraph form for solving infinite-horizon constrained optimization problems [59].

## 4 Solving with Reinforcement Learning

In the previous section, we reformulated the minimum-cost reach-avoid problem by constructing an augmented system and used its reach value function (11) in a new constrained optimization problem (13) over the cost upper-bound $z_0$. In this section, we propose Reachability Constrained Proximal Policy Optimization (RC-PPO), a two-phase RL-based method for solving (13) (see Figure 1).

### 4.1 Phase 1: Learn $z$-conditioned policy and value function

In the first step, we learn the optimal policy $\pi$ and value function $\tilde{V}_{\hat{g}}^{\pi}$, as functions of the cost upper-bound $z_0$, using RL. To do so, we consider the policy gradient framework [64]. However, since the policy gradient requires a stochastic policy in the case of deterministic dynamics [65], we consider an analog of the developments made in the previous section but for the case of a stochastic policy. To this end, we redefine the reach value function $\tilde{V}_{\hat{g}}^{\pi}$ using a similar Bellman equation under a stochastic policy as follows.

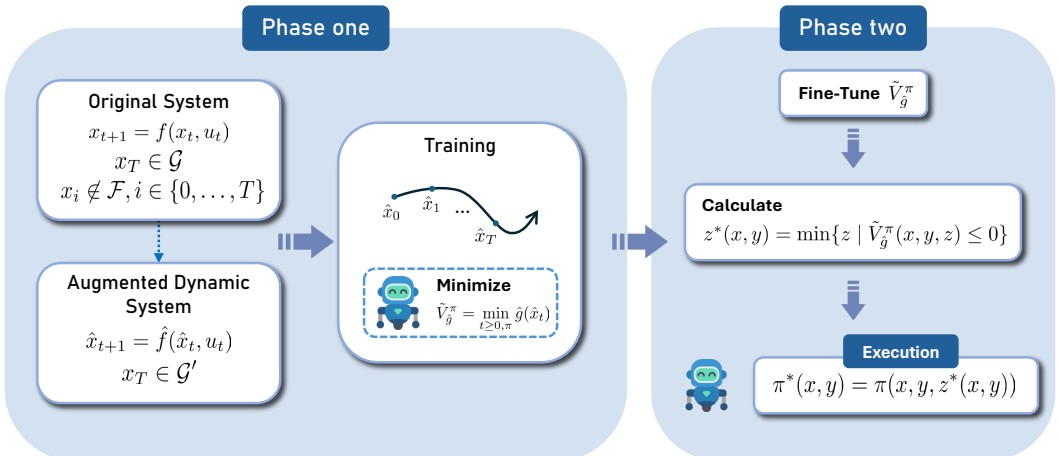

Figure 1: **Summary of the RC-PPO algorithm.** In phase one, the original dynamic system is transformed into the augmented dynamic system defined in (7). Then RL is used to optimize value function $\tilde{V}_{\hat{g}}^{\pi}$ and learn a stochastic policy $\pi$. In phase two, we fine-tune $\tilde{V}_{\hat{g}}^{\pi}$ on a deterministic version of $\pi$ and compute the optimal upper-bound $z^*$ to obtain the optimal deterministic policy $\pi^*$.

**Definition 1** (Stochastic Reachability Bellman Equation). Given function $\hat{g}$ in (9), a stochastic policy $\pi$, and initial conditions $x_0 \in \mathcal{X}, z_0 \in \mathbb{R}$, the stochastic reach value function $\tilde{V}_{\hat{g}}^{\pi}$ is defined as the solution to the following stochastic reachability Bellman equation (SRBE):

$$\tilde{V}_{\hat{g}}^{\pi}(\hat{x}_t) = \mathbb{E}_{\tau \sim \pi}[\min\{\hat{g}(\hat{x}_t), \tilde{V}_{\hat{g}}^{\pi}(\hat{x}_{t+1})\}] \quad \forall t \geq 0, \tag{14}$$

where $\hat{x}_0 = (x_0, y_0, z_0)$ with $y_0 = \mathbb{I}_{x_0 \in \mathcal{F}}$.

For this stochastic Bellman equation, the Q function [66] is defined as

$$\tilde{Q}_{\hat{g}}^{\pi}(\hat{x}_t, u_t) = \min\{\hat{g}(\hat{x}_t), \tilde{V}_g(\hat{x}_{t+1})\}. \tag{15}$$

We next define the dynamics of our problem with stochastic policy below.

**Definition 2** (Reachability Markov Decision Process). The Reachability Markov Decision Process is defined on the augmented dynamic in Equation (7) with an added absorbing state $s_0$. We define the transition function $f'_r$ with the absorbing state as

$$f'_r(\hat{x}, u) = \begin{cases} \hat{f}(\hat{x}, u), & \text{if } \tilde{V}_{\hat{g}}^{\pi}(\hat{x}) > \hat{g}(\hat{f}(\hat{x}, u)), \\ s_0, & \text{if } \tilde{V}_{\hat{g}}^{\pi}(\hat{x}) \leq \hat{g}(\hat{f}(\hat{x}, u)). \end{cases} \tag{16}$$

Denote by $d'_{\pi}(\hat{x})$ the stationary distribution under stochastic policy $\pi$ starting at $\hat{x} \in \mathcal{X} \times \{-1, 1\} \times \mathbb{R}$.

We now derive a policy gradient theorem for the Reachability MDP in Definition 2 which yields an almost identical expression for the policy gradient.

**Theorem 2.** (Policy Gradient Theorem) For policy $\pi_\theta$ parameterized by $\theta$, the gradient of the policy value function $\tilde{V}_{\hat{g}}^{\pi_\theta}$ satisfies

$$\nabla_\theta \tilde{V}_{\hat{g}}^{\pi_\theta}(\hat{x}) \propto \mathbb{E}_{\hat{x}' \sim d'_{\pi}(\hat{x}), u \sim \pi_\theta} \left[ \tilde{Q}^{\pi_\theta}(\hat{x}', u) \nabla_\theta \ln \pi_\theta(u \mid \hat{x}') \right], \tag{17}$$

under the stationary distribution $d'_{\pi}(\hat{x})$ for Reachability MDP in Definition 2

The proof of this new policy gradient theorem (Theorem 2) follows the proof of the normal policy gradient theorem [66], differing only in the expression of the stationary distribution. We provide the proof in Appendix D.3.

Since the stationary distribution $d'_{\pi}(\hat{x})$ in Definition 2 is hard to simulate during the learning process, we instead consider the stationary distribution under the original augmented dynamic system. Note that Definition 1 does not induce a contraction map, which harms performance. To fix this, we apply the same trick as [58] by introducing an additional discount factor $\gamma$ into the Bellman equation (12):

$$\tilde{V}_{\hat{g}}^{\pi}(\hat{x}_t) = (1 - \gamma)\hat{g}(\hat{x}_t) + \gamma \mathbb{E}_{\hat{x}_{t+1} \sim \tau}[\min\{\hat{g}(\hat{x}_t), \tilde{V}_{\hat{g}}^{\pi}(\hat{x}_{t+1})\}]. \tag{18}$$

This provides us with a contraction map (proved in [58]) and we leave the discussion of choosing $\gamma$ in Appendix C. The Q-function corresponding to (18) is then given as

$$\tilde{Q}_{\hat{g}}^{\pi}(\hat{x}_t, u_t) = (1 - \gamma)\hat{g}(\hat{x}_t) + \gamma \min\{\hat{g}(\hat{x}_t), \tilde{V}_{\hat{g}}^{\pi}(\hat{x}_{t+1})\}. \tag{19}$$

Following proximal policy optimization (PPO) [67], we use generalized advantage estimation (GAE) [68] to compute a variance-reduced advantage function $\tilde{A}_{\hat{g}}^{\pi} = \tilde{Q}_{\hat{g}}^{\pi} - \tilde{V}_{\hat{g}}^{\pi}$ for the policy gradient (Theorem 2) using the $\lambda$-return [66]. We refer to Appendix A for the definition of $\hat{A}_{\hat{g}}^{\pi(\text{GAE})}$ and denote the loss function when $\theta = \theta_l$ as

$$\mathcal{J}_{\pi}(\theta) = \mathbb{E}_{\hat{x}, u \sim \pi_{\theta_l}} \left[ \overline{A^{\pi_{\theta_l}}}(\hat{x}, u) \right], \tag{20}$$

$$\overline{A^{\pi_{\theta_l}}}(\hat{x}, u) = \max \left( -\frac{\pi_{\theta}(u \mid \hat{x})}{\pi_{\theta_l}(u \mid \hat{x})} \hat{A}_{\hat{g}}^{\pi_{\theta_l}(GAE)}(\hat{x}, u), \ \text{CLIP}\left(\epsilon, -\hat{A}_{\hat{g}}^{\pi_{\theta_l}(GAE)}(\hat{x}, u)\right) \right). \tag{21}$$

We wish to obtain the optimal policy $\pi$ and the value function $\tilde{V}_{\hat{g}}^{\pi_\theta}$ conditioned on $z_0$. Hence, at the beginning of each rollout, we uniformly sample $z_0$ within a user-specified range $[z_{\min}, z_{\max}]$. Since the optimal $z_0$ is the cumulative cost of the policy that solves the minimum-cost reach-avoid problem, $z_{\min}$ and $z_{\max}$ are user-specified bounds on the optimal cost. In particular, when the cost-function is bounded and the optimal cost is non-negative, we can choose $z_{\min}$ to be some negative number and $z_{\max}$ to be the maximum possible discounted cost.

### 4.2 Phase 2: Solving for the optimal $z$

In the second phase, we first compute a deterministic version $\pi^*$ of the stochastic policy $\pi$ from phase 1 by taking the mode. Next, we fine-tune $V_{\hat{g}}^{\pi}$ based on the now deterministic $\pi^*$ to obtain $\tilde{V}_{\hat{g}}^*$.

Given any state $x$, the final policy is then obtained by solving for the optimal cost upper-bound $z^*$ from Equation (13), which is a 1D root-finding problem and can be easily solved using bisection. Note that Equation (13) must be solved online for $z^*$ at each state $x$. Alternatively, to avoid performing bisection online, we can instead learn the map $(x, y) \mapsto z^*$ *offline* using regression with randomly sampled $(x, y)$ pairs and $z^*$ labels obtained from bisection offline.

We provide a convergence proof of an actor-critic version of our method without the GAE estimator in Appendix E.

## 5 Experiments

**Baselines** We consider two categories of RL baselines. The first is goal-conditioned reinforcement learning which focuses on goal-reaching but does not consider minimization of the cost. For this category, we consider the Contrastive Reinforcement Learning (CRL) [33] method. We also compare against safe RL methods that solve CMDPs. As the minimum-cost reach-avoid problem (3) cannot be posed as a CMDP, we reformulate (3) into the following *surrogate* CMDP:

$$\min_{\pi} \quad \mathbb{E}_{x_t, u_t \sim d_\pi} \sum_t \left[ -\gamma^t r(x_t, u_t) \right] \tag{22a}$$

$$\text{s.t.} \quad \mathbb{E}_{x_t, u_t \sim d_\pi} \sum_t \left[ \gamma^t \mathbb{1}_{x_t \in \mathcal{F}} \times C_{\text{fail}} \right] \leq 0, \tag{22b}$$

$$\mathbb{E}_{x_t, u_t \sim d_\pi} \sum_t \left[ \gamma^t c(x_t, u_t) \right] \leq \mathcal{X}_{\text{threshold}}. \tag{22c}$$

where the reward $r$ incentivies goal-reaching, $C_{\text{fail}}$ is a term balancing two constraint terms, and $\mathcal{X}_{\text{threshold}}$ is a hyperparameter on the cumulative cost. For this category, we consider the CPPO [48] and RESPO [60]. Note that RESPO also incorporates reachability analysis to adapt the Lagrange multipliers for each constraint term. We implement the above CMDP-based baselines with three different choices of $\mathcal{X}_{\text{thresholds}}$: $\mathcal{X}_{\text{L}}$, $\mathcal{X}_{\text{M}}$ and $\mathcal{X}_{\text{H}}$. For RESPO, we found $\mathcal{X}_{\text{M}}$ to outperform both $\mathcal{X}_{\text{L}}$ and $\mathcal{X}_{\text{H}}$ and thus only report results for $\mathcal{X}_{\text{M}}$.

We also consider the static Lagrangian multiplier case. In this setting, the reward function becomes $r(x_t) - \beta(\mathbb{1}_{x_t \in \mathcal{F}} \times C_{\text{fail}} + c(x_t, u_t))$ for a constant Lagrange multiplier $\beta$. We consider two different levels of $\beta$ ($\beta_{\text{L}}, \beta_{\text{H}}$) in our experiments, resulting in the baselines PPO_$\beta_{\text{L}}$, PPO_$\beta_{\text{H}}$, SAC_$\beta_{\text{L}}$, SAC_$\beta_{\text{H}}$. More details are provided in Appendix F.

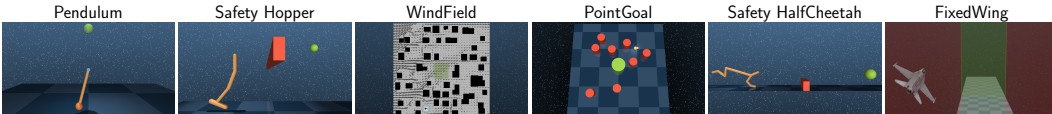

Figure 2: **Illustrations of the benchmark tasks.** In each picture, red denotes the unsafe region to be avoided, while green denotes the goal region to be reached.

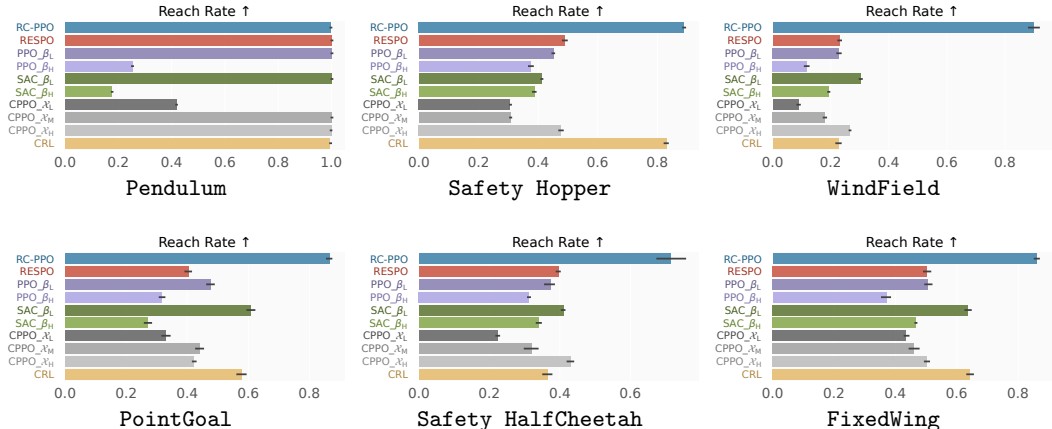

Figure 3: **Reach rates under the sparse reward setting.** RC-PPO consistently achieves the highest reach rates in all benchmark tasks. Error bars denote the standard error.

**Benchmarks** We compare RC-PPO with baseline methods on several minimum-cost reach-avoid environments. We consider an inverted pendulum (`Pendulum`), an environment from Safety Gym [69] (`PointGoal`) and two custom environments from MuJoCo [70], (`Safety Hopper`, `Safety HalfCheetah`) with added hazard regions and goal regions. We also consider a 3D quadrotor navigation task in a simulated wind field for an urban environment [71, 72] (`WindField`) and an Fixed-Wing avoid task from [59] with an additional goal region (`FixedWing`). More details on the benchmark can be found in Appendix G.

**Evaluation Metrics** Since the goal of RC-PPO is minimizing cost consumption while reaching goal without entering the unsafe region $\mathcal{F}$. We evaluate algorithm performance based on (i) reach rate, (ii) cost. The **reach rate** is the ratio of trajectories that enter goal region $\mathcal{G}$ without violating safety along the trajectory. The **cost** denotes the cumulative cost over the trajectory $\sum_{k=0}^{T} c(x_k, \pi(x_k))$.

## 5.1 Sparse Reward Setting

We first compare our algorithm with other baseline algorithms under a sparse reward setting (Figure 3). In all environments, the reach rate for the baseline algorithms is very low. Also, there is a general trend between the reach rate and the Lagrangian coefficient. CPPO_$\mathcal{X}_L$, PPO_$\beta_H$ and SAC_$\beta_H$ have higher Lagrangian coefficients which lead to a lower reach rate.

## 5.2 Comparison under Reward Shaping

Reward shaping is a common method that can be used to improve the performance of RL algorithms, especially in the sparse reward setting [73, 74]. To see whether the same conclusions still hold even in the presence of reward shaping, we retrain the baseline methods but with reward shaping using a distance function-based potential function (see Appendix F for more details).

The results in Figure 4 demonstrate that RC-PPO remains competitive against the best baseline algorithms in reach rate while achieving significantly lower cumulative costs. The baseline methods (PPO_$\beta_H$, SAC_$\beta_H$, CPPO_$\mathcal{X}_L$) fail to achieve a high reach rate due to the large weights placed on minimizing the cumulative cost. CRL can reach the goal for simpler environments (`Pendulum`) but struggles with more complex environments. However, since goal-conditioned methods do not consider minimize cumulative cost, it achieves a higher cumulative cost relative to other methods.

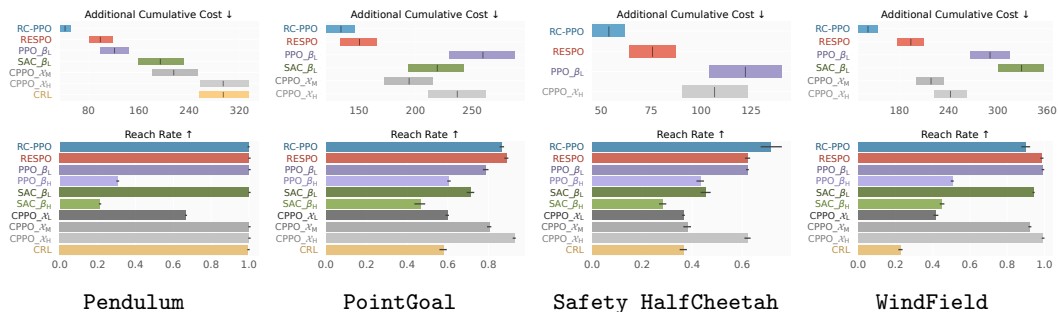

Figure 4: **Cumulative cost (IQM) and reach rates under reward shaping on four selected benchmarks.** RC-PPO achieves significantly lower cumulative costs while retaining comparable reach rates even when compared with baseline methods that use reward shaping.

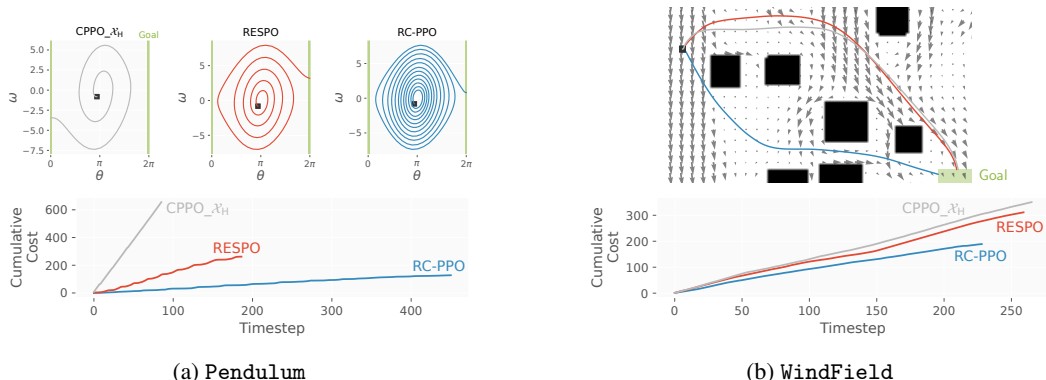

(a) `Pendulum`                              (b) `WindField`

Figure 5: **Trajectory comparisons.** On `Pendulum`, RC-PPO learns to perform an extensive energy pumping strategy to reach the goal upright position (green line), resulting in vastly lower cumulative energy. On `WindField`, RC-PPO takes advantage instead of fighting against the wind field, resulting in a faster trajectory to the goal region (green box) that uses lower cumulative energy. The start of the trajectory is marked by ■.

Other baselines focus more on goal-reaching tasks while putting less emphasis on the cost part. As a result, they suffer from higher costs than RC-PPO. We can also observe that RESPO achieves lower cumulative cost compared to CPPO_$\mathcal{X}_M$ which shares the same $\mathcal{X}_{\text{threshold}}$. This is due to RESPO making use of reachability analysis to better satisfy constraints.

To see how RC-PPO achieves lower cumulative costs, we visualize the resulting trajectories for `Pendulum` and `WindField` in Figure 5. For `Pendulum`, we see that RC-PPO learns to perform energy pumping to reach the goal in more time but with a smaller cumulative cost. The optimal behavior is opposite in the case of `WindField`, which contains an additional constant term in the cost to model the energy draw of quadcopters (see Appendix G). Here, we see that RC-PPO takes advantage of the wind at the beginning by moving *downwind*, arriving at the goal faster and with less cumulative cost.

We also visualize the learned RC-PPO policy for different values of $z$ on the `Pendulum` benchmark (see Appendix H.2). For small values of $z$, the policy learns to minimize the cost, but at the expense of not reaching the goal. For large values of $z$, the policy reaches the goal quickly but at the expense of a large cost. The optimal $z_{\text{opt}}$ found using the learned value function $\tilde{V}_{\hat{g}}^{\pi_\theta}$ finds the $z$ that minimizes the cumulative cost but is still able to reach the goal.

### 5.3 Optimal solution of minimum-cost reach-avoid cannot be obtained using CMDP

Though the previous subsections show the performance benefits of RC-PPO over existing methods, this may be due to badly chosen hyperparameters for the baseline methods, particularly in the formulation of the *surrogate* CMDP (22). We thus pose the following question: **Can CMDP methods perform well under the right parameters of the surrogate CMDP problem** (22) **?**.

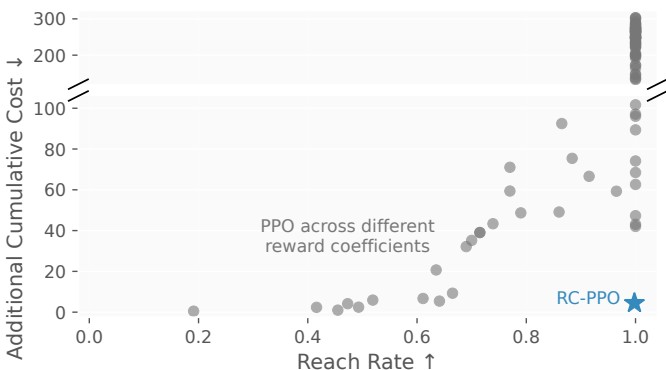

Figure 6: **Pareto front of PPO across different reward coefficients.** RC-PPO outperforms the *entire Pareto front* of what can be achieved by varying the reward function coefficients of the surrogate CMDP problem when solved using PPO.

**Empirical Study.** To answer this question, we first perform an extensive grid search over both the different coefficients in (22) and the static Lagrange multiplier for PPO (see Appendix H.3) and plot the result in Figure 6. RC-PPO outperforms the entire Pareto front formed from this grid search, providing experimental evidence that the performance improvements of RC-PPO stem from having a better problem formulation as opposed to badly chosen hyperparameters for the baselines.

**Theoretical Analysis on Simple Example.** To complement the empirical study, we provide an example of a simple minimum-cost reach-avoid problem where we prove that no choice of hyperparameter leads to the optimal solution in Appendix I.

### 5.4 Robustness to Noise

Finally, we investigate the robustness to varying levels of control noise in Appendix H.4. Even with the added noise, RC-PPO achieves the lowest cumulative cost while maintaining a comparable reach rate to other methods.

## 6 Conclusion and Limitations

We have proposed RC-PPO, a novel reinforcement learning algorithm for solving minimum-cost reach-avoid problems. We have demonstrated the strong capabilities of RC-PPO over prior methods in solving a multitude of challenging benchmark problems, where RC-PPO learns policies that match the reach rates of existing methods while achieving significantly lower cumulative costs.

However, it should be noted that RC-PPO is not without limitations. First, the use of augmented dynamics enables folding the safety constraints within the goal specifications through an additional binary state variable. While this reduces the complexity of the resulting algorithm, it also means that two policies that are both unable to reach the goal can have the same value $\tilde{V}_{g'}^{\pi}$ even if one is unsafe, which can be undesirable. Next, the theoretical developments of RC-PPO are dependent on the assumptions of deterministic dynamics, which can be quite restrictive as it precludes the use of commonly used techniques for real-world deployment such as domain randomization. We acknowledge these limitations and leave resolving these challenges as future work.

## Acknowledgments and Disclosure of Funding

This work was partly supported by the National Science Foundation (NSF) CAREER Award #CCF-2238030, the MIT Lincoln Lab, and the MIT-DSTA program. Any opinions, findings, conclusions, or recommendations expressed in this publication are those of the authors and don't necessarily reflect the views of the sponsors.

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

## A  GAE estimator Definition

Note, however, that the definition of return (19) is *different* from the original definition and hence will result in a different equation for the GAE.

To simplify the form of the GAE, we first define a "reduction" function $\phi^{(n)} : \mathbb{R}^n \to \mathbb{R}$ that applies itself recursively to its $n$ arguments, i.e.,

$$\phi^{(n)}(x_1, x_2, \ldots, x_n) := \phi^{(1)}\left(x_1,\ \phi^{(n-1)}(x_2, \ldots, x_n)\right)$$

where

$$\phi^{(1)}(x, y) := (1 - \gamma)x + \gamma \min\{x, y\}.$$

The $k$-step advantage function $\hat{A}_{\hat{g}}^{\pi(k)}$ can then be written as

$$\hat{A}_{\hat{g}}^{\pi(k)}(\hat{x}_t) = \phi^{(k)}\left(\hat{g}(\hat{x}_t), \ldots, \hat{g}(\hat{x}_{t+k-1}), \tilde{V}_{\hat{g}}^{\pi}(\hat{x}_{t+k})\right) - \tilde{V}_g^{\pi}(\hat{x}_t).$$

We can then construct the GAE $\hat{A}_{\hat{g}}^{\pi(\mathrm{GAE})}$ as the $\lambda^k$-weighted sum over the $k$-step advantage functions $\hat{A}_{\hat{g}}^{\pi(k)}$: Overall, the GAE estimator can be described as

$$\hat{A}_{\hat{g}}^{\pi(\mathrm{GAE})}(\hat{x}_t) = \frac{1}{1 - \lambda} \sum_{k=1}^{\infty} \lambda^k \hat{A}_{\hat{g}}^{\pi(k)}(\hat{x}_t).$$

## B  Equivalence of Problem 3 and Problem 13

The equivalence between the transformed Problem 13 and the original minimum-cost reach-avoid Problem 3 can be shown in the following sequence of optimization problems that yield the **exact same solution** if feasible:

$$\min_{\pi,T} \quad \sum_{k=0}^{T} c(x_k, \pi(x_k)) \quad \text{s.t} \quad g(x_T) \leq 0, \quad \max_{k=0,\ldots,T} h(x_k) \leq 0 \tag{23}$$

$$= \min_{\pi,T} \quad \sum_{k=0}^{T} c(x_k, \pi(x_k)) \quad \text{s.t} \quad g(x_T) \leq 0, \quad I_{(\max_{k=0,\ldots,T} h(x_k)>0)} \leq 0 \tag{24}$$

$$= \min_{z_0,\pi,T} \quad z_0 \quad \text{s.t.} \quad \sum_{k=0}^{T} c(x_k, \pi(x_k)) \leq z_0, \quad g(x_T) \leq 0, \quad I_{(\max_{k=0,\ldots,T} h(x_k)>0)} \leq 0 \tag{25}$$

$$= \min_{z_0,\pi,T} \quad z_0 \quad \text{s.t.} \quad \max\left(\sum_{k=0}^{T} c(x_k, \pi(x_k)) - z_0,\ g(x_T),\ I_{(\max_{k=0,\ldots,T} h(x_k)>0)}\right) \leq 0 \tag{26}$$

$$= \min_{z_0,\pi,T} \quad z_0 \quad \text{s.t.} \quad \hat{g}(\hat{x}_T) \leq 0 \tag{27}$$

$$= \min_{z_0} \quad z_0 \quad \text{s.t.} \quad \min_{\pi} \min_{T} \hat{g}(\hat{x}_T) \leq 0 \tag{28}$$

$$= \min_{z_0} \quad z_0 \quad \text{s.t.} \quad \min_{\pi} \tilde{V}_{\hat{g}}^{\pi}(\hat{x}_T) \leq 0 \tag{29}$$

This shows that the minimum-cost reach-avoid Problem 3 is equivalent to the formulation we solve in this work (29), which is Problem 13 in the paper. The formulation of RC-PPO solves (29) and thus also solves Problem 3 because they are equivalent.

## C  Optimal Reach Value Function

As shown in (18), we introduce an additional discount factor into the estimation of $\tilde{V}_{\hat{g}}^{\pi}$. It will incur imprecision on the calculation of $\tilde{V}_{\hat{g}}^{\pi}$ defined in Definition 1. In this section, we show that for a large enough discount factor $\gamma < 1$, we could reach unbiased $\tilde{z}$ in phase two of **RC-PPO**.

**Theorem 3.** We denote $\max_{\hat{x}\in\hat{\mathcal{X}}}\{\hat{g}(\hat{x})\} = G_{max}$ and maximal episode length $T_{max}$. If there exists a positive value $\epsilon$ where

$$\hat{g}(\hat{x}) < 0 \Rightarrow \hat{g}(\hat{x}) < -\epsilon.$$

Then for any $\frac{\gamma^{T_{max}}}{1-\gamma^{T_{max}}} > \frac{G_{max}}{\epsilon}$, for any deterministic policy $\pi$ satisfies 18. If there exists a trajectory under given policy $\pi$ leading to the extended goal region $\hat{\mathcal{G}}$. We have

$$\tilde{V}_{\hat{g}}^{\pi}(\hat{x}) < 0.$$

The proof for Theorem 3 is provided in Appendix D.4.

# D Proofs

## D.1 Proof for Theorem 1

*Proof.* We separately consider three elements in augmented state $(x_T, y_T, z_T)$. First, note that 3b holds if and only if $x_T \in \mathcal{G}$. For the second element $y$, from the definition of the augmented dynamics 7, it holds that

$$y_T = \max_{i\in\{0,...,T\}} \mathbb{I}_{x_i\in\mathcal{F}} \tag{30}$$

As a result 3c holds if and only if $y_T = -1$. For the third element $z$, note that $z_T = z_0 - \sum_{k=0}^{T-1} c(x_k, u(x_k))$. Hence, $z_T \geq 0$ if and only if $z_0 \geq \sum_{k=0}^{T-1} c(x_k, u(x_k))$. $\square$

## D.2 Proof for Property 12

*Proof.* From Definition 12, we know

$$
\begin{aligned}
\tilde{V}_{\hat{g}}^{\pi}(\hat{x}) &= \min_{t\in\mathbb{N}} \hat{g}(\hat{x}_t \mid \hat{x}_0 = \hat{x}) \\
&= \min\{\hat{g}(\hat{x}), \min_{t\in\mathbb{N}^+} \hat{g}(\hat{x}_t \mid \hat{x}_0 = \hat{x})\} \\
&= \min\{\hat{g}(\hat{x}), \tilde{V}_g^{\pi}(\hat{x}_{t+1})\}
\end{aligned}
$$

$\square$

## D.3 Proof for Theorem 2

We first derive the state value function in a recursive form similar as [66]

*Proof.*

$$
\begin{aligned}
\nabla_\theta \tilde{V}_{\hat{g}}^{\pi_\theta}(\hat{x}) =& \nabla_\theta \left( \sum_{u\in\mathcal{U}} \pi_\theta(u \mid \hat{x}) \tilde{Q}_{\hat{g}}^{\pi_\theta}(\hat{x}, u) \right) \\
=& \sum_{u\in\mathcal{U}} \left( \nabla_\theta \pi_\theta(u \mid \hat{x}) \tilde{Q}_{\hat{g}}^{\pi_\theta}(, u) + \pi_\theta(u \mid \hat{x}) \nabla_\theta \tilde{Q}_{\hat{g}}^{\pi_\theta}(\hat{x}, u) \right) \\
=& \sum_{u\in\mathcal{U}} \left( \nabla_\theta \pi_\theta(u \mid \hat{x}) \tilde{Q}_{\hat{g}}^{\pi_\theta}(\hat{x}, u) \right. \\
& \left. + \pi_\theta(u \mid \hat{x}) \nabla_\theta \min\{\hat{g}(\hat{x}), \tilde{V}_g^{\pi}(\hat{x}')\} \right) \\
=& \sum_{u\in\mathcal{U}} \left( \nabla_\theta \pi_\theta(u \mid \hat{x}) \tilde{Q}_{\hat{g}}^{\pi_\theta}(\hat{x}, u) \right. \\
& \left. + \pi_\theta(u \mid \hat{x}) \mathbb{1}_{\hat{g}(\hat{x})>\tilde{V}_g^{\pi_\theta}(\hat{x}')} \nabla_\theta \tilde{V}_g^{\pi_\theta}(\hat{x}') \right)
\end{aligned}
$$

where $\hat{x}' = \hat{f}(\hat{x}, u)$

Next, we consider unrolling $\tilde{V}_g^{\pi_\theta}(\hat{x}')$ under Reachability MDP in Definition 2. We define $\Pr(\hat{x} \to \hat{x}^\dagger, k, \pi_\theta)$ as the probability of transitioning from state $\hat{x}$ to $\hat{x}^\dagger$ in $k$ steps under policy $\pi_\theta$ in 2. Note that $\mathbb{1}_{\hat{g}(\hat{x}) > \tilde{V}_g^{\pi_\theta}(\hat{x}')}$ is absorbed using the absorbing state in 2. Then we can get

$$\nabla_\theta \tilde{V}_{\hat{g}}^{\pi_\theta}(\hat{x}) = \sum_{\hat{x}^\dagger \in \hat{\mathcal{X}}} \left( \sum_{k=0}^{\infty} \Pr\left(\hat{x} \to \hat{x}^\dagger, k, \pi\right) \right) \sum_{u \in \mathcal{U}} \nabla \pi_\theta(u \mid \hat{x}^\dagger) \tilde{Q}_{\hat{g}}^{\pi_\theta}(\hat{x}^\dagger, u)$$

$$\propto \mathbb{E}_{\hat{x}' \sim d'_\pi(\hat{x}), u \sim \pi_\theta} \left[ \tilde{Q}^{\pi_\theta}\left(\hat{x}', u\right) \nabla_\theta \ln \pi_\theta\left(u \mid \hat{x}'\right) \right]$$

$\square$

### D.4 Proof for Theorem 3

*Proof.* Consider trajectory $\{\hat{x}_0, \ldots, \hat{x}_T\}$ where $\hat{x}_T \in \hat{\mathcal{G}}$. We consider the worst-case scenario where $\hat{g}(\hat{x}_t) = g_{max}$ for $t \in \{0, \ldots, T-1\}$. Then

$$\begin{aligned}
\tilde{V}^\pi(\hat{x}_0) &= (1-\gamma)\hat{g}(\hat{x}_0) + \gamma \min\{\tilde{V}^\pi(\hat{x}_1), \hat{g}(\hat{x}_1)\} \\
&\leq (1-\gamma)g_{max} + \gamma \tilde{V}^\pi(\hat{x}_1) \\
&\leq (1-\gamma)g_{max} + \gamma((1-\gamma)g_{max} + \gamma \tilde{V}^\pi(\hat{x}_1)) \\
&\leq \sum_{i=0}^{T-1} \gamma^i (1-\gamma)g_{max} + \gamma^T \tilde{V}^\pi(\hat{x}_T) \\
&< (1-\gamma^T)g_{max} + \gamma^T \epsilon \\
&< 0
\end{aligned}$$

$\square$

## E Convergence Guarantee on an Actor-Critic Version of Our Method

In this section, we provide the convergence proof of phase one of our method under the actor-critic framework. Notice that similar to Bellman equation (18) for $\tilde{V}_{\hat{g}}^\pi$. We could also derive the Bellman equation for $\tilde{Q}_{\hat{g}}^\pi$ as

$$\tilde{Q}_{\hat{g}}^\pi(\hat{x}_t, u_t) = (1-\gamma)\hat{g}(\hat{x}_t) + \gamma \mathbb{E}_{\hat{x}_{t+1} \sim \tau, u_{t+1} \sim \pi}[\min\{\hat{g}(\hat{x}_t), \tilde{Q}_g^\pi(\hat{x}_{t+1}, u_{t+1})\}]$$

Next, we show our method under the actor-critic framework without GAE estimator in Algorithm 1

---

**Algorithm 1** RC-PPO (Actor Critic)

---

**Require:** Initial policy parameter $\theta_0$, Q function parameter $\omega_0$, horizon T, convex projection operator $\Gamma_\Theta$, and value function learning rate $\beta_1(k)$, policy learning rate $\beta_2(k)$
1: **for** k = 0, 1, ... **do**
2:     **for** t = 0 to T-1 **do**
3:         Sample trajectories $\tau_t : \{\hat{x}_t, u_t, \hat{x}_{t+1}\}$
4:         **Critic update:** $\omega_{k+1} = \omega_k - \beta_1(k)\nabla_\omega \tilde{Q}_{\hat{g}}(\hat{x}_t, u_t; \omega_k) \cdot$
5:         $\left[ \tilde{Q}_{\hat{g}}(\hat{x}_t, u_t; \omega_k) - \left( (1-\gamma)\hat{g}(\hat{x}_t) + \gamma \min\left\{\hat{g}(\hat{x}_t), \tilde{Q}_{\hat{g}}(\hat{x}_{t+1}, u_{t+1}; \omega_k)\right\} \right) \right]$
6:         **Actor Update:** $\theta_{k+1} = \Gamma_\Theta\left( \theta_k + \beta_2(k)\tilde{Q}_{\hat{g}}(\hat{x}_t, u_t; \omega_k) \nabla_\theta \log \pi_\theta(u_t \mid \hat{x}_t) \right)$
7:     **end for**
8: **end for**
9: **return** parameter $\theta, \omega$

---

In this algorithm, the $\Gamma_\Theta(\theta)$ operator projects a vector $\theta \in \mathbb{R}^k$ to the closest point in a compact and convex set $\Theta \subset \mathbb{R}^k$, i.e., $\Gamma_\Theta(\theta) = \arg\min_{\theta' \in \Theta} \|\theta' - \theta\|^2$.

Next, we provide the convergence analysis for Algorithm 1 under the following assumptions.

**Assumption 1.** (Step Sizes) The step size schedules $\{\beta_1(k)\}$ and $\{\beta_2(k)\}$ have below properties:

$$\sum_k \beta_1(k) = \sum_k \beta_2(k) = \infty$$

$$\sum_k \beta_1(k)^2, \sum_k \beta_2(k)^2 < \infty$$

$$\beta_2(k) = o(\beta_1(k))$$

**Assumption 2.** (Differentiability and and Lipschitz Continuity) For any state and action pair $(\hat{x}, u)$, $\tilde{Q}_{\hat{g}}(\hat{x}, u; \omega)$ and $\pi(\hat{x}; \theta)$ are continuously differentiable in $\omega$ and $\theta$. Furthermore, for any state and action pair $(\hat{x}, u)$, $\nabla_\omega \tilde{Q}_{\hat{g}}(\hat{x}, u; \omega)$ and $\pi(\hat{x}; \theta)$ are Lipschitz function in $\omega$ and $\theta$.

Also, we assume that $\mathcal{X}$ and $\mathcal{U}$ are finite and bounded and the horizon $T$ is also bounded by $T_{\max}$, then the cost function $c$ can be bounded by $C_{\max}$ and $g$ can be bounded within $G_{\max}$. We can limit the space of cost upper bound $z \in [-G_{\max}, T \cdot C_{\max}]$ instead of $\mathbb{R}$. This is due to $\hat{g}(\hat{x}) = -z$ for $z \le -G_{\max}$. Next, we could do discretization on $[-G_{\max}, T \cdot C_{\max}]$ and cost function $c$ to make the augmented state set $\hat{\mathcal{X}}$ finite and bounded.

With the above assumptions, we can provide a convergence guarantee for Algocrithm 1.

**Theorem 4.** Under Assumptions 1 and 2, the policy update in Algorithm 1 converge almost surely to a locally optimal policy.

*Proof.* The proof follows from the proof of Theorem 2 in [75], differing only in whether an update exists for the Lagrangian multiplier. We provide a proof sketch as

- First, we prove that the critic parameter almost surely converges to a fixed point $\omega^*$.

This step is guaranteed by the assumption of finite and bounded state and action set and Assumption 1. The $\gamma$-contraction property of the following operator

$$\mathcal{B}[\tilde{Q}_{\hat{g}}(\hat{x}, u)] = (1 - \gamma)\hat{g}(\hat{x}) + \gamma \mathbb{E}_{\hat{x}' \sim \tau, u \sim \pi}[\min\{\hat{g}(\hat{x}), \tilde{Q}_g^\pi(\hat{x}', u)\}] \tag{31}$$

is also proved in Lemma B.1 in [75] to make sure the convergence of the first step.

- Second, due to the fast convergence of $\omega^*$, we can show policy paramter $\theta$ converge almost surely to a stationary point $\theta^*$ which can be further proved to be a locally optimal solution.

We refer to [75] for proof details.

$\square$

## F    Implementation Details of Algorithms

In this section, we will provide more details about CMDP-based baselines (different between optimization goal with multiple constraints) and other hyperparameter settings like $\mathcal{X}_{\text{threshold}}$.

### F.1    CMDP-based Baselines

In this section, we will clarify the optimization target for CPPO and RESPO under CMDP formulation of both hard and soft constraints. Recall that our formulation of CMDP is

$$\min_\pi \quad \mathbb{E}_{x_t, u_t \sim d_\pi} \sum_t \left[ -\gamma^t r(x_t, u_t) \right] \tag{32a}$$

$$\text{s.t.} \quad \mathbb{E}_{x_t, u_t \sim d_\pi} \sum_t \left[ \gamma^t \mathbb{1}_{x_t \in \mathcal{F}} \times C_{fail} \right] \le 0, \tag{32b}$$

$$\mathbb{E}_{x_t, u_t \sim d_\pi} \sum_t \left[ \gamma^t c(x_t, u_t) \right] \le \mathcal{X}_{\text{threshold}} \tag{32c}$$

We then denote

$$V_r^\pi(x_t) := \mathbb{E}_{x_t, u_t \sim d_\pi} \sum_t \left[ \gamma^t r(x_t, u_t) \right]$$

$$V_f^\pi(x_t) := \mathbb{E}_{x_t, u_t \sim d_\pi} \sum_t \left[ \gamma^t \mathbb{1}_{x_t \in \mathcal{F}} \times C_{cost} \right]$$

$$V_c^\pi(x_t) := \mathbb{E}_{x_t, u_t \sim d_\pi} \sum_t \left[ \gamma^t c(x_t, u_t) \right]$$

The optimization goal formulation for CPPO is as follows:

$$\min_\pi \max_\lambda \left( L(\pi, \lambda) = -V_r^\pi(x) + \lambda_c \cdot (V_c^\pi(x) - \mathcal{X}_{threshold}) + \lambda_f \cdot V_f^\pi(x) \right)$$

In this formulation, the soft constraint $V_c^\pi$ has the same priority as the hard constraint $V_f^\pi$. This leads to a potential imbalance between soft constraints and hard constraints. Instead, the optimization goal for RESPO is as follows:

$$\min_\pi \max_\lambda L(\pi, \lambda) = \left( -V_r^\pi(x) + \lambda_c \cdot (V_c^\pi(x) - \mathcal{X}_{threshold}) \right.$$
$$\left. + \lambda_f \cdot V_f^\pi(x) \right) \cdot (1 - p(x)) + p(x) \cdot V_f^\pi(x)$$

where $p(x)$ denotes the probability of entering the unsafe region $\mathcal{F}$ start from state $x$. It is called the reachability estimation function (REF). This formulation prioritizes the satisfaction of hard constraints but still suffers from balancing soft constraints and reward terms.

### F.2  Hyperparameters

We first clarify how we set proper $\mathcal{X}_{\text{threshold}}$ for each environment. First, we will run our method RC-PPO and calculate the average cost, we denote it as $c_{average}$. We set $\mathcal{X}_{\text{low}} = \frac{c_{average}}{10}$, $\mathcal{X}_{\text{medium}} = \frac{c_{average}}{3}$ and $\mathcal{X}_{\text{high}} = c_{average}$. For static lagrangian multiplier $\beta$, we set $\beta_{\text{lo}} = 0.1$ and $\beta_{\text{hi}} = 10$. Also, we set $C_{fail} = 20$ in every environment.

Note that CRL is an off-policy algorithm, while RC-PPO and other baselines are on-policy algorithms. We provide Table 1 showing hyperparameters for on-policy algorithms and Table 2 showing hyperparameters for off-policy algorithm (CRL).

### F.3  Implementation of the baselines

The implementation of the baseline follows their original implementations:

- RESPO: https://github.com/milanganai/milanganai.github.io/tree/main/NeurIPS2023/code (No license)
- CRL: `https://github.com/google-research/google-research/tree/master/contrastive_rl` (No License)

## G  Experiment Details

In this section, we provide more details about the benchmarks and the choice of reward function $r$, $g$, cost function $c$ and $C_{cost}$ in each environment. Under the sparse reward setting, we apply the following structure of reward design

$$r(x_t, u_t, x_{t+1}) = R_{goal} \times \mathbb{1}_{x_{t+1} \in \mathcal{G}}$$

where $R_{goal}$ is an constant. After doing reward shaping, we add an extra term $\gamma \phi(x_{t+1}) - \phi(x_t)$ and the reward becomes

$$r(x_t, u_t, x_{t+1}) = R_{goal} \times \mathbb{1}_{x_{t+1} \in \mathcal{G}} + \gamma \phi(x_{t+1}) - \phi(x_t)$$

where $\gamma$ denotes the discount factor.

Note that we set $R_{goal} = C_{cost} = 20$ in all the environments. Note that if there is a gap between $\max\{g(x) \mid g(x) < 0\}$, we could get unbiased $\tilde{z}$ during phase two of RC-PPO guaranteed by Theorem 3. To achieve better performance in phase two of RC-PPO, we set

$$g(x) = -300$$

for all $x \in \mathcal{G}$ to maintain such a gap. Also, we implement all the environments in Jax [76] for better scalability and parallelization.

Table 1: Hyperparameter Settings for On-policy Algorithms

| Hyperparameters for On-policy Algorithms | Values |
| --- | --- |
| **On-policy parameters** | |
| Network Architecture | MLP |
| Units per Hidden Layer | 256 |
| Numbers of Hidden Layers | 2 |
| Hidden Layer Activation Function | tanh |
| Entropy coefficient | Linear Decay 1e-2 $\rightarrow$ 0 |
| Optimizer | Adam |
| Discount factor $\gamma$ | 0.99 |
| GAE lambda parameter | 0.95 |
| Clip Ratio | 0.2 |
| Actor Learning rate | Linear Decay 3e-4 $\rightarrow$ 0 |
| Reward/Cost Critic Learning rate | Linear Decay 3e-4 $\rightarrow$ 0 |
| **RESPO specific parameters** | |
| REF Output Layer Activation Function | sigmoid |
| Lagrangian multiplier Output Layer Activation function | softplus |
| Lagrangian multiplier Learning rate | Linear Decay 5e-5 $\rightarrow$ 0 |
| REF Learning Rate | Linear Decay 1e-4 $\rightarrow$ 0 |
| **CPPO specific parameters** | |
| $K_P$ | 1 |
| $K_I$ | 1e-4 |
| $K_D$ | 1 |

Table 2: Hyperparameter Settings for Off-policy Algorithms

| Hyperparameters for Off-policy Algorithms | Values |
| --- | --- |
| **Off-policy parameters** | |
| Network Architecture | MLP |
| Units per Hidden Layer | 256 |
| Numbers of Hidden Layers | 2 |
| Hidden Layer Activation Function | tanh |
| Entropy target | -2 |
| Optimizer | Adam |
| Discount factor $\gamma$ | 0.99 |
| Actor Learning rate | Linear Decay 3e-4 $\rightarrow$ 0 |
| Critic Learning rate | Linear Decay 3e-4 $\rightarrow$ 0 |
| Actor Target Entropy | 0 |
| Replay Buffer Size | 1e6 transitions |
| Replay Batch Size | 256 |
| Train-Collect Interval | 16 |
| Target Smoothing Term | 0.005 |

### G.1 Pendulum

The Pendulum environment is taken from Gym [18] and the torque limit is set to be 1. The state space is given by $x = [\theta, \dot{\theta}]$ where $\theta \in [-\pi, \pi], \dot{\theta} \in [-8, 8]$. In this task, we do not consider unsafe regions and set

$$\mathcal{G} := \{[\theta, \dot{\theta}] \mid \theta \cdot (\theta + \dot{\theta} \cdot dt) < 0\}$$

where $dt = 0.05$ is the time interval during environment simulation. This is for preventing environment overshooting during simulation.

In the Pendulum environment, cost function $c$ is given by

$$c(x_t, u_t, x_{t+1}) = \begin{cases} 0 & \text{if} \quad \|u_t\| < 0.1 \\ 8\|u\|^2 & \text{if} \quad \|u_t\| \geq 0.1 \end{cases}$$

for better visualization of policies with different energy consumption. $g$ is given by

$$g(x) = \begin{cases} 100\theta^2 & \text{if} \quad x \notin \mathcal{G} \\ -300 & \text{if} \quad x \in \mathcal{G} \end{cases}$$

### G.2 Safety Hopper

The Safety Hopper environment is taken from Safety Mujoco, we add static obstacles in the environment to increase the difficulty of the task. We use $x$ to denote the x-axis position of the head of Hopper, $y$ to be the y-axis position of the head of Hopper. Then the goal region can be described as

$$\mathcal{G} := \{(x, y) \mid \|[x, y] - [2.0, 1.4]\| < 0.1\}$$

The unsafe set is described as

$$\mathcal{F} := \{(x, y) \mid 0.95 \leq x \leq 1.05, y \geq 1.3\}$$

We use $\tilde{x}^{thigh}, \tilde{x}^{leg}, \tilde{x}^{foot}$ to denote the angular velocity of the thigh, leg, foot hinge. The cost function is described as

$$c(x_t, u_t, x_{t+1}) = l(x_t^{thigh}, u_t^1) + l(x_t^{leg}, u_t^2) + l(x_t^{foot}, u_t^3)$$

where

$$l(a, b) = \begin{cases} 0 & \text{if} \quad \|a \cdot b\| < 0.4 \\ 0.15a^2 \cdot b^2 & \text{if} \quad \|a \cdot b\| > 0.4 \end{cases}$$

$g$ is given by

$$g(\tilde{x}) = \begin{cases} 100\sqrt{(x-2)^2 + 100(y-1.4)^2} - 40 & \text{if} \quad \tilde{x} \notin \mathcal{G} \\ -300 & \text{if} \quad \tilde{x} \in \mathcal{G} \end{cases}$$

### G.3 Safety HalfCheetah

The Safety HalfCheetah environment is taken from Safety Mujoco, we add static obstacles in the environment to increase the difficulty of the task. We use $x_{front}$ to denote the x-axis position of the front foot of Halfcheetah, $y_{front}$ to be the y-axis position of the back foot of Halfcheetah, $x_{back}$ to denote the x-axis position of the back foot of Halfcheetah, $y_{back}$ to be the y-axis position of the back foot of Halfcheetah, $x_{head}$ to denote the x-axis position of the head of Halfcheetah, $y_{head}$ to be the y-axis position of the head of Halfcheetah. Then the goal region can be described as

$$\mathcal{G} := \{(x_{head}, y_{head}) \mid \|[x_{head}, y_{head}] - [5.0, 0.0]\| < 0.2\}$$

The unsafe set is described as

$$\mathcal{F} := \{(x_{front}, y_{front}) \mid y_{front} < 0.25, 2.45 < x_{front} < 2.55\}$$
$$\cup \{(x_{back}, y_{back}) \mid y_{back} < 0.25, 2.45 < x_{back} < 2.55\}$$

The cost function is described as

$$c(x_t, u_t, x_{t+1}) = \|u_t\|^2$$

$g$ is given by

$$g(\tilde{x}) = \begin{cases} 100\sqrt{(x_{head} - 2)^2 + (y_{head} - 1.4)^2} - 20 & \text{if} \quad \tilde{x} \notin \mathcal{G} \\ -300 & \text{if} \quad \tilde{x} \in \mathcal{G} \end{cases}$$

### G.4 FixedWing

FixedWing environment is taken from [59] and we follow the same design of $\mathcal{F}$ as [59]. We denote the $x_{PE}$ as the eastward displacement of F16 with given state $x$. Then the goal region $\mathcal{G}$ is given by

$$\mathcal{G} := \{x \mid 1975 \leq x_{PE} \leq 2025\}$$

The cost $c$ is given by

$$c(x_t, u_t, x_{t+1}) = 4\|u_t/[1, 25, 25, 25]\|^2$$

and $g$ is given by

$$g(x) = \begin{cases} \frac{\|x_{PE} - 2000\| - 25}{4} & \text{if} \quad x \notin \mathcal{G} \\ -300 & \text{if} \quad x \in \mathcal{G} \end{cases}$$

### G.5 Quadrotor in Wind Field

We take quadrotor dynamics from crazyflies and wind field environments in the urban area from [71]. The wind field will disturb the quadrotor with extra movement on both $x$-axis and $y$-axis. There are static building obstacles in the environment and we treat them as the unsafe region $\mathcal{F}$. The goal for the quadrotor is to reach the mid-point of the city. We divide the whole city into four sections and train single policy on each of the sections. We use $x \in [-30, 30]$ to denote the x-axis position of quadrotor, $y \in [-30, 30]$ to be the y-axis position of quadrotor.

$$\mathcal{G} := \{(x, y) \mid \|[x, y]\| \leq 4\}$$

The cost $c$ is given by

$$c(x_t, u_t, x_{t+1}) = \frac{\|u_t\|^2}{2}$$

$g$ is given by

$$g(\tilde{x}) = \begin{cases} 10\sqrt{(x - x_{goal})^2 + 10(y - y_{goal})^2} - 40 & \text{if} \quad x \notin \mathcal{G} \\ -300 & \text{if} \quad x \in \mathcal{G} \end{cases}$$

### G.6 PointGoal

The PointGoal environment is taken from Safety Gym [69] We implement `PointGoal` environments in Jax. In Safety Gym environment, we do not perform reward-shaping and use the original reward defined in Safety Gym environments. In this case, the distance reward is set also to be 20 in order to align* with $C_{goal}$ and $C_{cost}$. Different from sampling outside the hazard region which is implemented in Safety Gym, we allow Point to be initialized within the hazard region. We use $x$ to denote the x-axis position of Point, $y$ to be the y-axis position of Point, $x_{goal}$ to denote the x-axis position of Goal, and $y_{goal}$ to denote the y-axis position of Goal. The goal region is given by

$$\mathcal{G} := \{(x, y) \mid \|[x, y] - [x_{goal}, y_{goal}]\| \leq 0.3\}$$

The cost $c$ is given by

$$c(x_t, u_t, x_{t+1}) = \frac{\|u_t\|^2}{2}$$

$g$ is given by

$$g(\tilde{x}) = \begin{cases} 100\sqrt{(x - x_{goal})^2 + (y - y_{goal})^2} - 30 & \text{if} \quad x \notin \mathcal{G} \\ -300 & \text{if} \quad x \in \mathcal{G} \end{cases}$$

### G.7 Experiment Harware

We run all our experiments on a computer with CPU AMD Ryzen Threadripper 3970X 32-Core Processor and with 4 GPUs of RTX3090. It takes at most 4 hours to train on every environment.

# H  Additional Experiment Results

We put additional experiment results in this section.

## H.1  Additional Cumulative Cost and Reach Rates

We show the cumulative cost and reach rates of the final converged policies for additional environments (`F16` and `Safety Hopper`) in Figure 7.

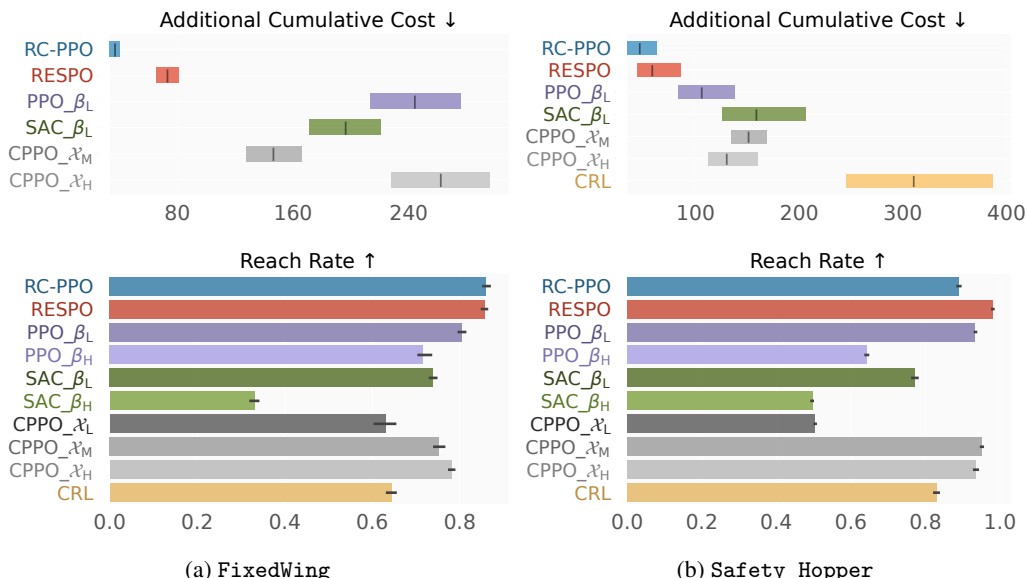

(a) `FixedWing`  (b) `Safety Hopper`

Figure 7: Cumulative cost and reach rates of the final converged policies.

## H.2  Visualization of learned policy for different $z$

To obtain better intuition for how the learned policy depends on $z$, we rollout the policy choices of $z_0$ in the `Pendulum` environment and visualize the results in Figure 8.

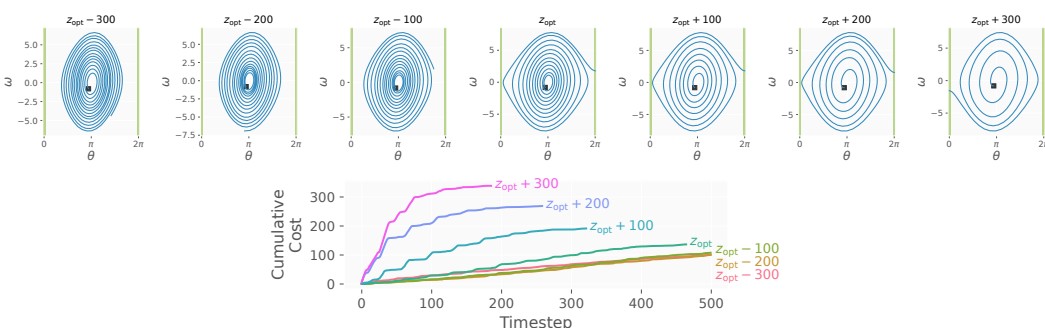

Figure 8: Learned RC-PPO policy for different $z$ on `Pendulum`. For a smaller cost lower-bound $z$, cost minimization is prioritized at the expense of not reaching the goal. For a larger cost lower-bound $z$, the goal is reached using a large cumulative cost. Performing rootfinding to solve for the optimal $z_{\mathrm{opt}}$ *automatically* finds the policy that minimizes cumulative costs while still reaching the goal.

| Algorithm | Reach Rate | + Small Noise | + Large Noise |
|---|---|---|---|
| RCPPO | 1.00 | 1.00 | 1.00 |
| RESPO | 1.00 | 1.00 | 1.00 |
| PPO $\beta_L$ | 1.00 | 1.00 | 1.00 |
| PPO $\beta_H$ | 0.31 | 0.38 | 0.34 |
| SAC $\beta_L$ | 1.00 | 1.00 | 1.00 |
| SAC $\beta_H$ | 0.21 | 0.37 | 0.20 |
| CPPO $\mathcal{X}_L$ | 0.67 | 0.65 | 0.65 |
| CPPO $\mathcal{X}_M$ | 1.00 | 1.00 | 1.00 |
| CPPO $\mathcal{X}_H$ | 1.00 | 1.00 | 0.99 |
| CRL | 1.00 | 1.00 | 1.00 |

Table 3: Reach rate of final converged policies with different levels of noise to the output control

| Algorithm | Additional Cumulative Cost | + Small Noise | + Large Noise |
|---|---|---|---|
| RCPPO | 35.3 | 41.4 | 132.9 |
| RESPO | 92.0 | 93.6 | 179.2 |
| PPO $\beta_L$ | 97.7 | 98.6 | 150.2 |
| SAC $\beta_L$ | 156.3 | 157.6 | 270.5 |
| CPPO $\mathcal{X}_M$ | 223.2 | 220.5 | 209.0 |
| CPPO $\mathcal{X}_H$ | 212.7 | 299.8 | 298.4 |
| CRL | 228.3 | 229.1 | 261.1 |

Table 4: Additional cumulative cost of final converged policies with different levels of noise to the output control

### H.3 Grid search

We perform an extensive grid search over different reward coefficients for the baseline PPO method and plotted the Pareto front across the reach rate and cost in Figure 6. The reward we use is

$$r = R_{goal} \times \mathbb{1}_{x \in \mathcal{G}} - P_{goal} \times \mathbb{1}_{x \notin \mathcal{G}} - \beta c(x, u).$$

and we search over the Cartesian product of $R_{goal} = \{2, 20, 200, 2000, 20000\}, P_{goal} = \{1, 10, 100, 1000, 10000\}, \beta = \{0.1, 1, 10\}$.

### H.4 Performance with external noise

We also performed additional experiments to illustrate the performance of RC-PPO under a changing environment. Specifically, we add uniform noise to the output of the learned policy and see what happens in the `Pendulum` environment.

We first compare the reach rates of the different methods in Table 3. In this environment, we see that noise does not affect the reach rate too much.

Next, we look at how the cumulative cost changes with noise by comparing methods with a near 100% reach rate in Table 4. Unsurprisingly, larger amounts of noise reduce the performance of almost all policies. Even with the added noise, RC-PPO uses the least cumulative cost compared to all other methods.

## I   Discussion on Limitation of CMDP-Based Algorithms

In this section, we will use an example to illustrate further why CMDP-based algorithms won't solve the minimum-cost reach-avoid problem optimally compared with our method. We focus on two parts of CMDP formulation:

- Weight coefficient assigned to different objectives
- Threshold assigned to each constraint

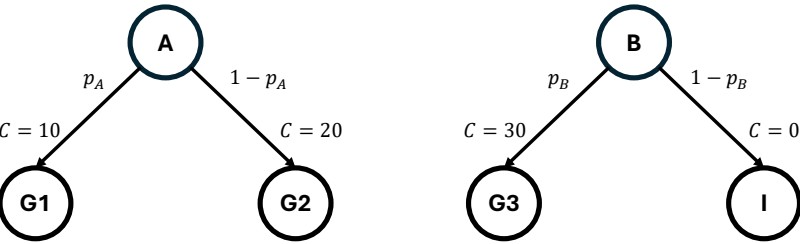

Figure 9: Minimum-cost reach-avoid example to illustrate the limitation of CMDP-based formulation.

Consider the minimum-cost reach-avoid problem shown in Figure 9, where we use $C$ to denote the cost. States $A$ and $B$ are two initial states with the same initial distribution probability. State $G_1$, $G_2$, and $G_3$ are three goal states. State $I$ is the absorbing state (non-goal). We use $p_A$ and $p_B$ to denote the policy parameter, which represents the probability of choosing *left* action on state $A$ and $B$ separately.

The optimal policy for this minimum-cost reach-avoid problem is to take the left action from both $A$ and $B$, i.e., $p_A = p_B = 1$, which gives an expected cost of

$$0.5 \cdot 10 + 0.5 \cdot 30 = 20$$

To convert this into a multi-objective problem, we introduce a reward that incentivizes reaching the goal as follows (we use $R$ to denote reward):

$$R(A, G_1) = 10, \ R(A, G_2) = 20, \ R(B, G_3) = 20, \ R(A, I) = 0 \tag{33}$$

This results in the following multi-objective optimization problem:

$$\min_{p_A, p_B \in [0,1]} \quad (-R, C) \tag{34}$$

### I.1 Weight assignment

We first consider solving multi-objective optimization Problem 34 by assigning weights on different objectives. We introduce $w \geq 0$, giving

$$\min_{p_A, p_B \in [0,1]} \quad -R + wC \tag{35}$$

Solving the scalarized Problem 34 gives us the following solution as a function of $w$:

$$p_A = \mathbb{1}_{(w \geq 1)}, \quad p_B = \mathbb{1}_{(w \leq \frac{2}{3})} \tag{36}$$

Notice that the *true* optimal solution of $p_A = p_B = 1$ is NOT an optimal solution to the original minimum-cost reach-avoid problem shown in Figure 9 under any $w$.

Hence, **the optimal solution of the surrogate multi-objective problem can be suboptimal for the original minimum-cost reach-avoid problem under any weight coefficients.**

Of course, this is just one choice of reward function where the optimal solution of the minimum-cost reach-avoid problem cannot be recovered. Given knowledge of the optimal policy, we can construct the reward such that the multi-objective optimization problem does include the optimal policy as a solution. However, this is impossible to do if we do not have prior knowledge of the optimal policy, as is typically the case.

### I.2 Threshold assignment

Next, we consider solving multi-objective optimization Problem 34 by assigning a threshold $\mathcal{X}\_{\text{thresh}}$ on the cost constraint:

$$0.5(10p_A + 20(1 - p_A)) + 0.5(30p_B) \leq \mathcal{X}_{\text{thresh}} \tag{37}$$

The optimal solution to this CMDP can be solved to be

$$p_A = 0, \quad p_B = \frac{\mathcal{X}_{\text{thres}} - 10}{15}. \tag{38}$$

However, **the true optimal solution of $p_A = p_B = 1$ is NOT an optimal solution to the CMDP.**
To see this, taking $\mathcal{X}\_\text{thresh} = 20$, the real optimal solution $p_A = p_B = 1$ gives a reward of $R = 15$, but the CMDP solution $p_A = 0, p_B = \frac{20-10}{15} = \frac{2}{3}$ gives $R = 23.33 > 15$. Moreover, any uniform scaling of the rewards or costs does not change the solution.

We can "fix" this problem if we choose the rewards to be high only along the optimal solution $p_A = p_B = 1$, but this requires knowledge of the optimal solution beforehand and is not feasible for all problems.

Another way to "fix" this problem is if we consider a "per-state" cost threshold, e.g.,

$$10p_A + 20(1 - p_A) \leq \mathcal{X}_A, \qquad 10p_B + 20(1 - p_B) \leq \mathcal{X}_B \tag{39}$$

Choosing exactly the cost of the optimal policy, i.e., $\mathcal{X}_A = 10$ and $\mathcal{X}_B \geq 30$, also recovers the optimal solution of $p_A = p_B = 1$. This now requires knowing the smallest cost to reach the goal for every state, which is difficult to do beforehand and not feasible. On the other hand, RC-PPO does exactly this in the second phase when optimizing for $z_0$. We can thus interpret **RC-PPO as automatically solving for the best cost threshold to use as a constraint for every initial state.**

## J   Broader impact

Our proposed algorithm solves an important problem that is widely applicable to many different real-world tasks including robotics, autonomous driving, and drone delivery. Solving this brings us one step closer to more feasible deployment of these robots in real life. However, the proposed algorithm requires GPU training resources, which could contribute to increased energy usage.

