# OpenReview forum: "Solving Minimum-Cost Reach Avoid using Reinforcement Learning"
_NeurIPS.cc/2024/Conference — NeurIPS 2024 poster_

### Official Review · Reviewer_M8co · 2024-07-05

**Soundness:** 2
**Presentation:** 2
**Contribution:** 2
**Rating:** 6
**Confidence:** 4

**Summary:**

The papers coinsiders a reach-avoid problem under cost minimization objective, where a CMDP has to be solved under the goal and avoid constraints (the final state has to be in the certain set, and the trajectories must avoid an unsafe region). The authors derive an optimal control problem under these constraints and show how to solve it in the RL setting. The results show an improvement in terms of the optimized objective in comparison with method that include the goal objective in the optimization cost.

**Strengths:**

The paper delivers well the proposed solution and the state augmentation technique that the authors propose seem to be crucial to the solution. The algorithm does outperform the competitors, where the goal is included in the objective.

**Weaknesses:**

1. The state augmentation is not a novel technique in constrained RL, (see references below) albeit the application here is different.
1. I am not fully convinced that this is actually an important sub-problem in RL to solve. Can the authors elaborate why we should consider this problem in more detail? For instance, can we formulate safety gym benchmarks as this?
1. Some of the design choices are not fully clear, why the authors minimize the initial state z_0 with the constraint z_t >=0. Wouldn't it be easier to track the accumulated cost and minimize the terminal state?
1. I am not fully convinced that there are no simpler methods to solve this problem. For example, for the baselines, the authors reshape the reward by giving a modest reward for goal reaching. I’d like to see experiments with heavy penalization of not reaching the goal (for example, a scale from -10 to -10000) and different rewards for reaching the goal (not just one 20).


* [Sootla et al 22] Sootla, Aivar, et al. "Sauté rl: Almost surely safe reinforcement learning using state augmentation." International Conference on Machine Learning. PMLR, 2022.

* [Jiang et al 23] Jiang, Hao, et al. "Solving Richly Constrained Reinforcement Learning through State Augmentation and Reward Penalties." arXiv preprint arXiv:2301.11592 (2023).

* [Jiang et al 24] Jiang, Hao, et al. "Reward Penalties on Augmented States for Solving Richly Constrained RL Effectively." Proceedings of the AAAI Conference on Artificial Intelligence. Vol. 38. No. 18. 2024.

**Questions:**

1. I appreciate it’s a personal preference, but I don’t see the need to formulate the optimal control problem as cost minimization. Furthermore, in safe RL the costs typically refer to constraints.
2. Line 33 “Moreover, the use of nonlinear numerical optimization may result in poor solutions that lie in suboptimal local minima [14].” How does the paper address this issue? Wouldn’t any hard problem have the same problem?
3. Line 38 “However, posing the reach constraint as a reward then makes it difficult to  optimize for the cumulative cost at the same time.”. I think adding a penalty for not reaching the goal will address this issue. Recall that in RL the rewards need not to be differentiable.
4. Line 45. “However, the choice of  this fixed threshold becomes key: too small and the problem is not feasible, destabilizing the training  process. Too large, and the resulting policy will simply ignore the cumulative cost.” - I am not sure this statement is necessary. Problem statement is a bit of an art and the threshold is chosen on the problem design level. Sometimes it is simply given.

**Limitations:**

limitations are discussed

---

> ### Author Rebuttal · Authors · 2024-08-04
>
> >## The paper considers ... where a CMDP has to be solved
>
> We want to clarify **the minimum-cost reach-avoid problem is NOT a CMDP [1]** (e.g., as used in [2,3,4]), since it is NOT in the following form:
> $$
> \max_\pi \quad \mathbb{E}\left[ \sum_{k=0}^\infty r_k \right],\quad \textrm{s.t.} \quad \mathbb{E}\left[ \sum_{k=0}^\infty d \right] \leq c_\max.
> $$
>
> ---
> >## State augmentation is not a novel technique, albeit the application is different.
>
> **Thank you for the references to state augmentation, we will include them and discuss the relationship with our work in the final version. We use state augmentation differently to the referenced works**. We emphasize that the novelty is in the framework for solving minimum-cost reach-avoid problems. We _DO NOT_ claim to be the first to propose the _idea_ of state augmentation. Moreover, while [2-4] use the augmented state to satisfy CMDP constraints by modifying the reward function but keeping the same value function, we use the augmented state to enforce the "upper-bound property" (L145) and apply it to the reachability Bellman equation (7).
>
> ---
> >## Why is minimum-cost reach-avoid an important problem?
>
> **Minimum-cost reach-avoid (MCRA) is an important real-world problem, especially for \*climate change\*, but RL existing methods do not \*directly\* solve the problem.** In addition to "energy-efficient autonomous driving" (L20) and "spacecraft and quadrotors" (L23) mentioned in the introduction, we highlight the additional use cases of plasma fusion [5] (reach desired current, minimize total risk of plasma disruption) and voltage control [6] (reach voltage level, minimize load shedding amount).
>
> **Examples of MCRA in RL benchmarks:** Lunar Lander (reach goal, avoid crash, minimize thrusters), Reacher (reach goal, minimize control effort), PointGoal (reach goal, avoid unsafe region, minimize time).
>
> Since existing RL methods do not directly solve the minimum-cost reach-avoid problem, a surrogate problem needs to be manually designed. We show in Figure 4 that these methods achieve _suboptimal_ cumulative costs due to the use of this surrogate problem. This motivates us to construct a method that can solve MCRA _directly_.
>
> ---
> >## Can simpler methods solve this problem with the right reward function coefficients?
>
> **Unfortunately, no choice reward function coefficients match the performance of our RC-PPO even after an extensive search over the coefficients**. As suggested, we have added a penalty for not reaching the goal and performed an _extensive_ grid search over the coefficients of the reward function.
>
> **We plot the results in Fig. 2 in the response pdf (see caption for grid search details)**. Different choices of the coefficients trade between reach rate and cost, forming a Pareto front. However, even the best point is sub-optimal compared to our RC-PPO.
>
> ---
> >## Problem statement is a bit of an art. Sometimes it is simply given.
>
> **We agree that, given a minimum-cost reach-avoid problem, constructing a surrogate problem that yields a good solution to the original problem is hard.** If a (C)MDP is given then it can be solved directly. However, if a minimum-cost reach-avoid problem is given, it makes more sense to solve the problem _directly_ instead of relying on the "art" of reward function design to convert it to a (C)MDP. Moreover, the answer above shows that sometimes _no_ choice of coefficients gives the optimal solution.
>
> ---
> >## Why minimize the z_0 with the constraint z_t >=0?
>
> We DO NOT impose the constraint z_t >=0, as the question implies.
>
> The challenge of solving _constrained_ optimization problems is in handling the interplay between the _objective_ and the _constraint_. If it were only the _objective_, then unconstrained optimization methods can be used. Similarly, if there were only _constraints_, then reachability can be used.
>
> Our strategy in this work belongs to the latter approach, where we convert the cumulative cost _objective_ $\sum_{t=0}^{T-1} c$ into a cumulative-cost upper-bound _constraint_ (L145)
> $$
> \sum_{t=0}^{T-1} c(x_t, u_t) \leq z_0.
> $$
> This gives us a problem with only three constraints, which we can solve using reachability.
>
> If all constraints are satisfied (i.e., the goal is reached on the augmented system) for some $z_0$, then the cumulative cost is less than $z_0$ AND the other constraints are satisfied, which means the original constrained problem (1) can be solved with an objective of less than $z_0$. Consequently, the smallest $z_0$ that still satisfies all the constraints "thus corresponds to the minimum-cost policy that still satisfies the reach-avoid constraints" (L164).
>
> **Note**: As we write in Remark 1, our strategy here "can be interpreted as an epigraph reformulation of the minimum-cost reach-avoid problem (1)" (L168). This epigraph reformulation technique is standard optimization [7, p.134] but is not that well known.
>
> >## Would it be easier to track the accumulated cost and minimize the terminal state?
>
> **This is what we already do!** The reachability problem on the augmented system (4, 5) aims to keep the cumulative cost below $z_0$ and reach the goal at some terminal time.
>
> ---
> >## “nonlinear numerical optimization may result in ... suboptimal local minima [14].” How does the paper address this issue?
>
> **We do not claim to address the local minima issue for RL**. Rather, this sentence only serves to illustrates a weakness of nonlinear trajectory optimization when compared to RL which has been well-established previously in the literature [8,9].
>
> ---
> [1] Altman 1999, "CMDP"\
> [2] Aivar et al 2022, "Sauté rl..."\
> [3] Jiang et al 2023, "Solving Richly..."\
> [4] Jiang et al 2024, "Reward Penalties..."\
> [5] Wang et al 2024, "Active Disruption Avoidance ... Tokamak ..."\
> [6] Huang et al 2020, "Accelerated DeepRL ... Emergency Voltage Control"\
> [7] Boyd 2004, "Convex optimization"\
> [8] Suh 2022, "Do Differentiable..."\
> [9] Grandesso 2023, "CACTO..."

---

> ### Comment · Reviewer_M8co · 2024-08-11
>
> Thank you for a detailed response and additional experiments.
>
> I still have some concerns regarding the claims. I am very skeptical of the authors results since their formulations of reach-avoid problems do not solve the stated problem. This indicates the problem with the formulation or the hyperparameters. I don't think the authors made a significant effort to investigate why this occurs. However, the choices they made for the CMDP are not unreasonable, the authors detailed the experiments and the readers can see when the alternative fail. They also followed my suggestions which also didn't work well.
>
> Overall, I think the formulation is interesting and the authors have convinced me that it could be better to use directly than trying to formulate a CMPD and balance all the constraints.
>
> I am raising my score.

---

> ### Author Response · Authors · 2024-08-11
> **Additional Clarifications**
>
> Thank you for raising your score! Your suggested experiments have definitely helped to better show the fundamental limitations of solving the minimum-cost reach-avoid problem using a surrogate CMDP.
>
> We are happy to clarify any further questions you may have!
>
> ---
>
> >## I am very skeptical of the authors results since their formulations of reach-avoid problems do not solve the stated problem.
>
> Could you clarify what you mean here?
>
> Theoretically, **our formulation is an \*exact\* transformation of the minimum-cost reach-avoid problem (1).** This can be shown in the following sequence of optimization problems that yield the **exact same solution** if feasible:
>
> $$
> \begin{align}
> &\min_{\pi,T}\quad \sum_{k=0}^T c(x_k, \pi(x_k)) \quad \text{ s.t } \quad g(x_T) \leq 0, \quad \max_{k = 0, \dots, T} h(x_k) \leq 0 \tag{1} \\\\
> =& \min_{\pi,T}\quad \sum_{k=0}^T c(x_k, \pi(x_k)) \quad \text{ s.t } \quad g(x_T) \leq 0, \quad I_{(\max_{k = 0, \dots, T} h(x_k) > 0)} \leq 0 \tag{B} \\\\
> =& \min_{z_0, \pi, T}\quad z_0 \quad \text{ s.t. } \quad \sum_{k=0}^T c(x_k, \pi(x_k)) \leq z_0, \quad g(x_T) \leq 0,\quad I_{(\max_{k = 0, \dots, T} h(x_k) > 0)} \leq 0 \tag{C} \\\\
> =& \min_{z_0, \pi, T}\quad z_0 \quad \text{ s.t. } \quad \max\left( \sum_{k=0}^T c(x_k, \pi(x_k)) - z_0,\\; g(x_T),\\; I_{(\max_{k = 0, \dots, T} h(x_k) > 0)} \right) \leq 0 \tag{D} \\\\
> =& \min_{z_0, \pi, T}\quad z_0 \quad \text{ s.t. } \quad \hat{g}(\hat{x}_T) \leq 0 \tag{E} \\\\
> =& \min\_{z_0} \quad z_0 \quad \text{ s.t. } \quad \min\_\pi \min_T \hat{g}(\hat{x}_T) \leq 0 \tag{F} \\\\
> =& \min\_{z_0} \quad z_0 \quad \text{ s.t. } \quad \min\_\pi \tilde{V}\_{\hat{g}}^\pi (\hat{x}_T) \leq 0 \tag{G}
> \end{align}
> $$
>
> This shows that the minimum-cost reach-avoid problem (1) is **equivalent** to the formulation we solve in this work (G), which is (8) in the paper. The formulation of RC-PPO solves (G) and thus also solves (1) because they are equivalent.
>
> This new derivation will be included in the final version to improve clarity.

---

> > ### Comment · Reviewer_M8co · 2024-08-12
> >
> > I mean that the CMDP formulation, which is used as a baseline to compare, should solve the problem optimally, but the computational results imply that it doesn't. I cannot see specifically why and I am not sure that an explanation was provided.

---

> ### Author Response · Authors · 2024-08-13
>
> > ## Why does the CMDP formulation give suboptimal results?
>
> This is a very good question. **The optimal solution of the CMDP formulation is guaranteed to be suboptimal for the original minimum-cost reach-avoid problem for a wide range of problems**. The main culprit at play here is the fact that the cost threshold is a _fixed constant_. We illustrate this with the following example.
>
> ## Problem Setup
> Consider the following minimum-cost reach-avoid problem, where we use $C$ to denote the cost.
>
> - **Initial state distribution**: A ($p=0.5$), B ($p=0.5$)
> - **Goal states**: $G_1$, $G_2$, $G_3$
> - **(Non-goal) Absorbing state**: $I$
> - **Policy parameters**: $p_A$, $p_B \in [0, 1]$
>
> ```
>        ┌───┐
>   pA ┌─┤ A ├─┐ 1-pA
>      │ └───┘ │
> C=10 ▼       ▼ C=20
>    ┌──┐     ┌──┐
>    │G1│     │G2│
>    └──┘     └──┘
>        ┌───┐
>   pB ┌─┤ B ├─┐ 1-pB
>      │ └───┘ │
> C=30 ▼       ▼ C=0
>    ┌──┐     ┌─┐
>    │G3│     │I│
>    └──┘     └─┘
> ```
>
> The optimal policy to this minimum-cost reach-avoid problem is to take the _left_ action from both $A$ and $B$, i.e., $p_A = p_B = 1$, which gives an expected cost of
> $$
> 0.5 \cdot 10 + 0.5 \cdot 30 = 20 \tag{$\dagger$}
> $$
>
> ## CMDP Solution
> To convert this to a CMDP, we introduce a reward that incentivizes reaching the goal, and add a cost constraint with threshold $\mathcal{X}\_{\text{thresh}}$:
> $$
> 0.5(10p_A+20 (1-p_A)) + 0.5(30 p_B) \leq \mathcal{X}_{\text{thresh}}
> $$
> This gives the following CMDP (we use $R$ to denote reward):
> ```
>        ┌───┐
>   pA ┌─┤ A ├─┐ 1-pA
>      │ └───┘ │
> C=10 |       | C=20
> R=10 ▼       ▼ R=20
>    ┌──┐     ┌──┐
>    │G1│     │G2│
>    └──┘     └──┘
>        ┌───┐
>   pB ┌─┤ B ├─┐ 1-pB
>      │ └───┘ │
> C=30 |       | C=0
> R=20 ▼       ▼ R=0
>    ┌──┐     ┌─┐
>    │G3│     │I│
>    └──┘     └─┘
> ```
>
> The optimal solution to this CMDP can be solved to be
> $$
> p_A = 0, \quad p_B = \frac{\mathcal{X}_{\text{thres}} - 10}{15}. \tag{$\star$}
> $$
> However, **the \*true\* optimal solution of $p_A = p_B = 1$ is NOT an optimal solution to the CMDP ($\star$)**. To see this, taking $\mathcal{X}\_{\text{thresh}} = 20$ as in ($\dagger$), the real optimal solution $p_A = p_B = 1$ gives a reward of $R=15$, but the CMDP solution $p_A = 0, p_B = \frac{20 - 10}{15} = \frac{2}{3}$ in ($\star$) gives $R=23.33 > 15$. Moreover, any uniform scaling of the rewards or costs does not change the solution.
>
> ## Fixes
> We can "fix" this problem if we choose the rewards to be high only along the optimal solution $p_A = p_B = 1$, but this requires knowledge of the optimal solution beforehand and is not feasible for all problems.
>
> Another way to "fix" this problem is if we consider a "per-state" cost-threshold, e.g.,
> $$
> 10 p_A + 20(1-p_A) \leq \mathcal{X}_A, \qquad 10 p_B + 20(1-p_B) \leq \mathcal{X}_B
> $$
> Choosing exactly the cost of the optimal policy, i.e., $\mathcal{X}_A = 10$ and $\mathcal{X}_B \geq 30$, also recovers the optimal solution of $p_A = p_B =1$. This now requires knowing the smallest cost to reach the goal _for every state_, which is difficult to do beforehand and not feasible. On the other hand, **RC-PPO does exactly this in the second-phase when optimizing for $z_0$**. We can thus interpret RC-PPO as **automatically solving for the best cost-threshold to use as a constraint for every initial state**.
>
> ---
>
> We will include this explanation in the final version to improve clarity.

---

### Official Review · Reviewer_zsEG · 2024-07-10

**Soundness:** 3
**Presentation:** 3
**Contribution:** 3
**Rating:** 7
**Confidence:** 3

**Summary:**

The paper introduces RC-PPO, an RL algorithm designed to solve the minimum-cost reach-avoid problem by reformulating the optimization problem on an augmented system. The paper addresses the limitations of current RL that mostly solve surrogate problems to fit to the problem setting. Furthermore, a comprehensive analysis, including theoretical foundations, algorithmic details, and empirical validation is presented. Experimental comparison with respect to existing methods is also provided.

**Strengths:**

The paper (including the appendix) provides solid theoretical foundation, including proofs and detailed explanations of the key concepts and assumptions.
The paper is overall well-written, definitions and theorems are formulated clearly. The main thread of the paper can be followed.
The presented experimental results are well illustrated and generally the claims based on the results are comprehensible.
In addition to solid theoretical arguments, the appendix also provides vast implementational details and further experimental results.
The presented idea seems to offer an elegant solution to minimum-cost reach-avoid problems.

Overall a nice and insightful read.

**Weaknesses:**

(minor)
The presented experimental results are solid and illustrative, but could be extended.
The extra title of Figure 3 is slightly irritating.
Figure 3 and Figure 4 use different axis labeling logic.
Appendix G.2 is empty (- a formatting problem?).
There are typos, missing spaces, missing punctuation, repeated words and repeated phrases in the paper, e.g., L99, L186, L187, L193, L200, L206, L210+L225, L215, L232, L570.

**Questions:**

In Section 5, the paper claims that RC-PPO remains competitive which is based on Figure 6. Can you either elaborate how this claim can be made from Figure 6, or maybe it should be Figure 4?

**Limitations:**

Limitations are briefly discussed.

---

> ### Author Rebuttal · Authors · 2024-08-03
>
> > ### In Section 5, the paper claims that RC-PPO remains competitive which is based on Figure 6. Can you either elaborate how this claim can be made from Figure 6, or maybe it should be Figure 4?
>
> **You are correct, this should be Figure 4**.
>
> > ### The presented experimental results are solid and illustrative, but could be extended.
>
> **Thank you for the suggestion!** We have included additional comparisons to SAC (Figure 1 in the pdf), though the conclusions in the paper remain the same. We have also performed an extensive grid search over different reward coefficients for the baseline PPO method and plotted the Pareto front (Figure 2) across the reach rate and cost. Notably, RC-PPO vastly outperforms the entire Pareto front, demonstrating that methods that solve the surrogate problem yield suboptimal policies due to not solving for the minimum-cost reach-avoid problem _directly_.
>
> If you have any further suggestions on how the presented experimental results could be extended, we would _love_ to hear your thoughts!
>
> ---
> > #### The extra title of Figure 3 is slightly irritating. Figure 3 and Figure 4 use different axis labeling logic. Appendix G.2 is empty (- a formatting problem?). There are typos, missing spaces, missing punctuation, repeated words and repeated phrases in the paper, e.g., L99, L186, L187, L193, L200, L206, L210+L225, L215, L232, L570.
>
> **Thank you for the meticulous reading of the manuscript!** Unfortunately, NeurIPS does not allow for uploading a new version of the manuscript, but we will definitely include the changes in the final version.

---

> > ### Comment · Reviewer_zsEG · 2024-08-12
> >
> > I have read the individual rebuttals as well as the shared rebuttal and thank the authors for taking the time to answer.
> > I will maintain my current rating.

---

### Official Review · Reviewer_186A · 2024-07-11

**Soundness:** 2
**Presentation:** 3
**Contribution:** 2
**Rating:** 5
**Confidence:** 3

**Summary:**

This paper proposes Reach Constrained Proximal Policy Optimization which targets to solve the minimum-cost reach-avoid problem. The authors first convert the reach-avoid problem to a reach problem on an augmented system and use the corresponding reach value function to compute the optimal policy. Next, The authors use a novel two-step PPO-based RL-based framework to learn this value function and the corresponding optimal policy.

**Strengths:**

1.The studied problem is interesting
2.This paper is easy to follow

**Weaknesses:**

1.What is the motivation for using two-phase PPO to solve this problem? Providing performance comparisons with other RL algorithms, such as TD3 and SAC, will significantly strengthen the rationale behind proposing this approach.

2.To verify the statement in the abstract, "which leads to suboptimal policies that do not directly minimize the cumulative cost," the authors should compare the performance of RC-PPO with other multi-objective optimization algorithms.


3. Please reconsider whether it is reasonable to reformulate the minimum-cost reach-avoid problem by constructing an augmented system, as the limitation that "two policies that are both unable to reach the goal can have the same value even if one is unsafe" is undesirable.

**Questions:**

1.What is the motivation for using two-phase PPO to solve this problem? Providing performance comparisons with other RL algorithms, such as TD3 and SAC, will significantly strengthen the rationale behind proposing this approach.

2.To verify the statement in the abstract, "which leads to suboptimal policies that do not directly minimize the cumulative cost," the authors should compare the performance of RC-PPO with other multi-objective optimization algorithms.

**Limitations:**

3. Please reconsider whether it is reasonable to reformulate the minimum-cost reach-avoid problem by constructing an augmented system, as the limitation that "two policies that are both unable to reach the goal can have the same value even if one is unsafe" is undesirable.

---

> ### Author Rebuttal · Authors · 2024-08-03
>
> > ### What is the motivation of using two-phase PPO to solve this problem?
>
> **The minimum-cost reach-avoid problem (1) cannot be *exactly* framed into problem structures that existing methods are able to solve.**
> Our two-phase method _directly_ solves the minimum-cost reach-avoid problem (1). In comparison, alternate RL methods either:
> 1. _Only_ solve the reach-avoid problem without consideration of the cumulative cost.
> 2. Solve _unconstrained_ problems (e.g., PPO, SAC, TD3)
> 3. Solve problems with CMDP constraints that constrain the _expectation over initial states_ of the sum of some cost function (e.g., CPPO, RESPO)
>
> Since the minimum-cost reach-avoid problem (1) cannot be _exactly_ framed into any of the above problem types, this "prevents the straightforward application of existing RL methods to solve (1)" (L110).
>
> We introduce a two-phase method to extend reachability analysis to "additionally enable the minimization of the cumulative cost" (L126) on top of solving the traditional reach-avoid problem by constructing a new constraint called the "upper-bound property"  (L145) which enforces that "$z_0$ is an upper-bound on the total cost-to-come" (L144).
>
> 1. The first phase uses PPO to learn a _stochastic_ policy $\pi$ and corresponding value function $\tilde{V}_{\hat{g}}^\pi$ for different states $x$ and upper-bounds $z_0$.
> 2. However, the reachability analysis relies on a deterministic policy. Hence, in the second phase, we take a deterministic version of the learned stochastic policy $\pi$ and fine-tune the learned value function $\tilde{V}^{\pi}_{\hat{g}}$ on this deterministic policy.
> 3. Using the fine-tuned value function $\tilde{V}_{\hat{g}}^{\pi}$, we can then find the smallest $z_0$ that satisfies the constraint (8b), which will then "corresponds to the minimum-cost policy that satisfies the reach-avoid constraints" (L164).
>
> ---
> > ### Providing performance comparisons with other RL algorithms, such as TD3 and SAC, will significantly strengthen the rationale behind proposing this approach
>
> **Thank you for the suggestion!** Although we believe our approach to already be empirically well-motivated by the lower cumulative cost of our proposed method (Figure 4), **we have performed additional comparisons to SAC and PPO with more extensive reward shaping in the attached PDF**. The additional experiments lead to the same conclusion as in the main paper: because RC-PPO solves the original problem (1), it can achieve "significantly lower cumulative costs" (L265) while remaining "competitive against the best baseline algorithms in reach rate" (L264).
>
> ---
> > ### To verify the statement that solving the surrogate problem "leads to suboptimal policies that do not directly minimize the cumulative cost", the authors should compare the performance of RC-PPO with other multi-objective optimization algorithms
>
> As described in Section 5 (L231), we **already** compare RC-PPO against other methods that solve the surrogate CMDP problem (16) and find that "RC-PPO remains competitive against the best baseline algorithms in reach rate while achieving significantly lower cumulative costs" (L264, Figure 4). The reason why the baseline methods achieve a higher cumulative cost and are more suboptimal is _precisely_ because they are only able to solve a surrogate (16) instead of the true constrained optimization problem (1).
>
> Although we have compared against CPPO and RESPO, which are _single-policy_ multi-objective optimization algorithms [1,2], there also exist _multi-policy_ multi-objective optimization algorithms which aim to recover an approximation of the entire Pareto front [1,2]. We wish to emphasize that the minimum-cost reach-avoid problem does _NOT_ require solving for the entire Pareto front, as there is only a _single_ cumulative cost function albeit with multiple constraints. Even if the Pareto front was given, the optimal solution to (1) would still need to be found on the surface of the Pareto front, which is not a simple task.
>
> ---
> > ### The limitation that "two policies that are both unable to reach the goal can have the same value even if one is unsafe" is undesirable.
>
> **While this is undesirable, this is an artifact of our problem formulation (1)** where we pose a constrained optimization problem. Since the reach and avoid components are both constraints and hence "equal", any candidate that does not satisfy both constraints is _infeasible_ regardless of whether it is safe or not. We are happy that you agree this is an important future direction to investigate, which we have mentioned in the limitations section (Section L293). Since this is not an issue from the perspective of our current problem formulation (1), we believe it to be out of the scope of the current work, and "leave resolving these challenges as future work" (L297).
>
> ---
>
> [1] Policy Gradient Approaches for Multi–Objective Sequential Decision Making\
> [2] Empirical evaluation methods for multiobjective reinforcement learning algorithms

---

> > ### Comment · Reviewer_186A · 2024-08-12
> >
> > Thank you for a detailed response and additional experiments.
> > The problem studied by the authors can be transformed into a multi-objective optimization problem and solved using multi-objective algorithms. Although the authors have mathematically derived it into a problem that can be addressed using reinforcement learning and designed a reinforcement learning algorithm based on PPO, there are limitations in the mathematical tools used. Could you provide insights into the robustness and adaptability of the designed algorithm? Specifically, how does the algorithm perform when there are changes in the environment or when the weights assigned to reach-avoid and minimum-cost objectives vary?
> >
> > I am raising my score.

---

> > > ### Author Response · Authors · 2024-08-13
> > >
> > > Thank you for raising your score!
> > >
> > > Below,
> > > 1. We provide **new experiment results** on robustness: RC-PPO degrades gracefully as noise is introduced but _remains the lowest cost method among baseline method with high reach rates_.
> > > 2. Clarify the objective weights: RC-PPO _does not require weights by construction_, unlike the baseline methods we have compared against.
> > >
> > > **Please let us know if we have addressed all of your concerns!** We are happy to clarify any further questions you may have. All additional experiments and clarifications will be included in the final version.
> > >
> > >
> > > ---
> > > > ## How does the algorithm perform when there are changes in the environment
> > >
> > > Good question. Although not the focus of this work, **we have performed additional experiments to see what happens when the environment changes**. Specifically, we add uniform noise to the output of the learned policy and see what happens on the Pendulum environment.
> > >
> > > ### 1. Reach Rates
> > >
> > > We first compare the reach rates of the different methods. On this environment, we see that the presence of noise does not affect the reach rate too much.
> > >
> > > |   Name   |Reach Rate|&#124; + Small Noise|&#124; + Large Noise|
> > > |----------|---------:|------------:|------------:|
> > > |RC-PPO    |      1.00|         1.00|         1.00|
> > > |RESPO     |      1.00|         1.00|         1.00|
> > > |PPO $\beta_L$|      1.00|         1.00|         1.00|
> > > |PPO $\beta_H$|      0.31|         0.38|         0.34|
> > > |SAC $\beta_L$|      1.00|         1.00|         1.00|
> > > |SAC $\beta_H$|      0.21|         0.37|         0.20|
> > > |CPPO $\mathcal{X}_L$  |      0.67|         0.65|         0.65|
> > > |CPPO $\mathcal{X}_M$  |      1.00|         1.00|         1.00|
> > > |CPPO $\mathcal{X}_H$  |      1.00|         1.00|         0.99|
> > > |CRL       |      1.00|         1.00|         1.00|
> > >
> > > ### 2. Cumulative Cost
> > >
> > > Next, we look at how the cumulative cost changes with noise by _comparing methods with a near 100% reach rate_. Unsurprisingly, larger amounts of noise reduce the performance of almost all policies. Even with the added noise, **RC-PPO uses the least cumulative cost compared to all other methods.**
> > >
> > > |   Name   |Additional Cumulative Cost|&#124; + Small Noise|&#124; + Large Noise|
> > > |----------|-------------------------:|------------:|------------:|
> > > |RC-PPO    |                      35.3|         41.4|        132.9|
> > > |RESPO     |                      92.0|         93.6|        179.2|
> > > |PPO $\beta_L$ |                      97.7|         98.6|        150.2|
> > > |SAC $\beta_L$ |                     156.3|        157.6|        270.5|
> > > |CPPO $\mathcal{X}_M$  |                     223.2|        220.5|        209.0|
> > > |CPPO $\mathcal{X}_H$  |                     212.7|        299.8|        298.4|
> > > |CRL       |                     228.3|        229.1|        261.1|
> > >
> > >
> > > These additional experiments show that the **performance of RC-PPO degrades gracefully as more noise is added**, in line with existing RL methods.
> > >
> > > ---
> > > > ## How does the algorithm perform when the weights assigned to reach-avoid and minimum-cost objectives vary?
> > >
> > > **Our algorithm does NOT have any weight hyperparameters for the reach-avoid and the minimum-cost parts.** The motivation for our work is to solve the minimum-cost reach-avoid problem without the need to choose weights between the two. This is because the reach-avoid is a _constraint_ (always satisfy this), while the minimum-cost is an _objective_ (only minimize this if the reach-avoid constraint is satisfied).
> > >
> > > In comparison, the baseline methods DO need a choice of weights, since "all objectives are combined with a weighted sum" (L5). **We have compared with baseline methods that require such choice of weights in Section 5 and in the new pdf**, where RC-PPO outperforms all baselines for any choice of weights.

---

> > > > ### Comment · Reviewer_186A · 2024-08-13
> > > >
> > > > Thank you for a detailed response and additional experiments.
> > > > I decide to raise my score as part of my concerns have been solved.
> > > >
> > > > As your statement in the abstract " Instead, a surrogate problem is solved where all objectives are combined with a weighted
> > > > sum. However, this surrogate objective results in suboptimal policies that do not directly minimize the cumulative cost." I intuitively believe that the problem studied by the author can be transformed into a multi-objective problem, and appropriate parameters can be constructed in the multi-objective problem, such as the weight values between different objectives. Using non reinforcement learning methods to solve surrogate objective problems can also achieve the same performance as the author. I don't know how the author can refute my point of view experimentally or theoretically.

---

> ### Author Response · Authors · 2024-08-13
>
> Once again, thank you for raising your score!
>
> To answer your concerns about whether the minimum-cost reach-avoid problem can be solved using multi-objective methods, we have built a toy example to show that **our problem may not be solvable using multi-objective methods, depending on the choice of reward function**. Consequently, it is very difficult to construct surrogate reward functions that guarantee the optimality for the original minimum-cost reach-avoid problem in general.
>
> > ## Can the minimum-cost reach-avoid problem always be optimally solved as a surrogate multi-objective problem given proper weights between the different objectives?
>
> This is a very good question. **The optimal solution of the surrogate multi-objective problem can be suboptimal for the original minimum-cost reach-avoid problem given \*any\* choice of weights**. We illustrate this with the following example.
>
> ## Problem Setup
> Consider the following minimum-cost reach-avoid problem, where we use $C$ to denote the cost.
>
> - **Initial state distribution**: A ($p=0.5$), B ($p=0.5$)
> - **Goal states**: $G_1$, $G_2$, $G_3$
> - **(Non-goal) Absorbing state**: $I$
> - **Policy parameters**: $p_A$, $p_B \in [0, 1]$
>
> ```
>        ┌───┐
>   pA ┌─┤ A ├─┐ 1-pA
>      │ └───┘ │
> C=10 ▼       ▼ C=20
>    ┌──┐     ┌──┐
>    │G1│     │G2│
>    └──┘     └──┘
>        ┌───┐
>   pB ┌─┤ B ├─┐ 1-pB
>      │ └───┘ │
> C=30 ▼       ▼ C=0
>    ┌──┐     ┌─┐
>    │G3│     │I│
>    └──┘     └─┘
> ```
>
> The optimal policy for this minimum-cost reach-avoid problem is to take the _left_ action from both $A$ and $B$, i.e., $p_A = p_B = 1$, which gives an expected cost of
> $$
> 0.5 \cdot 10 + 0.5 \cdot 30 = 20
> $$
>
> ## Multi-objective Problem and Solution
> To convert this into a multi-objective problem, we introduce a reward that incentivizes reaching the goal as follows (we use $R$ to denote reward):
> ```
>        ┌───┐
>   pA ┌─┤ A ├─┐ 1-pA
>      │ └───┘ │
> C=10 │       │ C=20
> R=10 ▼       ▼ R=20
>    ┌──┐     ┌──┐
>    │G1│     │G2│
>    └──┘     └──┘
>        ┌───┐
>   pB ┌─┤ B ├─┐ 1-pB
>      │ └───┘ │
> C=30 │       │ C=0
> R=20 ▼       ▼ R=0
>    ┌──┐     ┌─┐
>    │G3│     │I│
>    └──┘     └─┘
> ```
> This results in the following multi-objective optimization problem:
> $$
> \min_{p_A, p_B \in [0, 1]} \quad (-R, C)
> $$
> To solve this multi-objective optimization problem, we employ scalarization and introduce a weight $w \geq 0$, giving
> $$
> \min_{p_A, p_B \in [0, 1]} \quad -R + w C  \tag{$\dagger$}
> $$
> Solving the scalarized problem ($\dagger$) gives us the following solution as a function of $w$:
> $$
> p_A = \mathbb{1}\_{(w \ge 1)}, \quad p_B = \mathbb{1}\_{(w \le \frac{2}{3})}
> $$
> Notice that **the \*true\* optimal solution of $p_A = p_B = 1$ is NOT an optimal solution to  ($\star$) under any $w$**.
>
> Hence, **the optimal solution of the surrogate multi-objective problem can be suboptimal for the original minimum-cost reach-avoid problem under any weight coefficients**.
>
> Of course, this is just one choice of reward function where the optimal solution of the minimum-cost reach-avoid problem cannot be recovered. Given knowledge of the optimal policy, we can construct the reward such that the multi-objective optimization problem ($\dagger$) does include the optimal policy as a solution. However, this is impossible to do if we do not have prior knowledge of the optimal policy, as is typically the case.
>
> In contrast, RC-PPO solves the minimum-cost reach-avoid problem directly. Hence, assuming the optimization is done exactly, the true solution will be found without the need for any reward design.
>
> ---
>
> We will include the above explanation in the final version to improve clarity.

---

### Author Rebuttal · Authors · 2024-08-03

We thank the reviewers for their valuable comments.

We are excited that the reviewers identified that we provide a _novel_ ($\color{#E24A33}{\textsf{M8co}}$) and _elegant_ ($\color{#348ABD}{\textsf{zsEG}}$) solution to the minimum-cost reach avoid problem that _improves upon the optimized objective_ ($\color{#988ED5}{\textsf{186A}}$) in comparison to existing methods, _addressing the limitations of current RL methods_ ($\color{#348ABD}{\textsf{zsEG}}$). Reviewers found our paper _well-written_ ($\color{#348ABD}{\textsf{zsEG}}$, $\color{#988ED5}{\textsf{186A}}$) and _easy to follow_ ($\color{#E24A33}{\textsf{M8co}}$, $\color{#348ABD}{\textsf{zsEG}}$). We believe that RC-PPO takes a significant step towards new RL algorithms that can solve minimum-cost reach-avoid problems by construction without the need for complex reward design.

---

# New Experiments

As _all_ reviewers have recognized our technical novelty, the primary criticism comes from an insufficient comparison to alternative RL methods ($\color{#E24A33}{\textsf{M8co}}$, $\color{#988ED5}{\textsf{186A}}$) and the necessity of RC-PPO for solving minimum-cost reach-avoid problems ($\color{#988ED5}{\textsf{186A}}$).
In the **attached PDF (below)**, we present
1. Additional comparisons with SAC on the six benchmark tasks in Fig 1 (as suggested by $\color{#E24A33}{\textsf{M8co}}$)
2. New comparison against an _extensive_ grid search over different reward coefficients in Fig 2 (as suggested by $\color{#988ED5}{\textsf{186A}}$).

In particular, the grid search over reward coefficients in Fig 2 forms a Pareto frontier, where different reward coefficients trade between the reach rate and the cumulative cost. **RC-PPO outperforms the _entire_ Pareto frontier, with _no single point on the Pareto frontier coming close to simultaneously achieving the high reach rates and low costs that RC-PPO achieves_**.

These results strengthen our argument that existing methods will lead to suboptimal policies since they do not _directly_ solve the minimum-cost reach-avoid problem. In contrast, RC-PPO is unique in that it solves the minimum-cost reach-avoid problem _directly_ and hence achieves lower cumulative costs.

We have tried our best to resolve all raised questions in the individual responses below. If you have any additional questions/comments/concerns, please let us know. We appreciate the reviewer's precious time in providing their valuable feedback.

---

**Please see the additional figures in the PDF below:**

---

> ### Author Response · Authors · 2024-08-07
> **Fig 2 Clarification: Blue Star marks the performance of RC-PPO**
>
> **Fig. 2 Clarification**: The $\textcolor{#348ABD}{\textsf{blue star}\unicode{x2605}}$ in the bottom right denotes **RC-PPO**. Each $\textcolor{#777777}{\textsf{grey point}}$ forming the Pareto front denotes one choice of reward function coefficient from the grid search.

---

### Author Response · Authors · 2024-08-14

As the author-reviewer discussion phase closes, we would like to thank all the reviewers for participating in the discussion and giving timely, constructive responses! The concerns that have been brought up have helped us to _clarify the motivation_, _improve the theoretical justification_ and provide _stronger empirical evidence_ of the benefits of solving minimum-cost reach-avoid problems using RC-PPO.

---

# Author-Reviewer Discussion Summary
We summarize the _major_ developments that have happened during the discussion below.

## New Experiments
- New comparison with SAC on all tasks (Rebuttal pdf, Fig 1)
    - SAC follows the same trends as the existing baseline methods in the paper, reaffirming our conclusions that RC-PPO outperforms other methods.
- Extensive grid search of reward function coefficients for baseline methods (Rebuttal pdf, Fig 2)
    - RC-PPO outperforms the _entire_ Pareto frontier. This provides experimental evidence that the surrogate CMDP which prior methods solve does not yield the optimal solution the minimum-cost reach-avoid problem. In contrast, RC-PPO solves the minimum-cost reach-avoid problem directly.
- Investigation into the robustness of the methods as noise is added into the environment (Reply to $\color{#988ED5}{\textsf{186A}}$)
    - Even with added noise, RC-PPO uses the least cumulative cost compared to all other methods. The performance of RC-PPO
degrades gracefully as more noise is added, in line with the other RL methods.

## New Theoretical Developments
- Proof that the formulation we eventually solve using reachability (8) is _equivalent_ to the minimum-cost reach-avoid problem (1). (Reply to $\color{#E24A33}{\textsf{M8co}}$)
- Proof that the CMDP formulation cannot optimally solve the minimum-cost reach-avoid problem for a wide range of setups. (Reply to $\color{#E24A33}{\textsf{M8co}}$)
- Proof that a surrogate multi-objective problem does not always optimally solve the minimum-cost reach-avoid problem. (Reply to $\color{#988ED5}{\textsf{186A}}$)

---

All developments, including clarifications in the replies to the reviewers that are not mentioned above, will be included in the final version.

**To the reviewers**: Thank you so much for participating in this productive author-reviewer discussion!

---

### Decision · Program_Chairs · 2024-09-25

**Decision:**

Accept (poster)

**Comment:**

The paper presents a method for solving Minimum-Cost Reach Avoid problems using RL. These are goal-driven sequential decision problems which include both constraints (avoiding set of states) together with minimization of cumulative cost. The authors consider the deterministic setting.

The paper is generally clearly written and easy to read.

The first proposed idea is to augment the state to include 'cumulative' information regarding the avoiding set of states (whether an avoiding state has been visited or not, so far) and the accumulated cost (actually z_0 minus the accumulated cost so far), where z_0 is some "upper bound" of the optimal cost at the initial state. Augmented state transition and goal states are also defined accordingly.

This idea is simple and it is difficult to see a priori how such a rewriting of the original problem can lead to any computational benefit. This reformulation also leads directly to the question of how to choose z_0.

The authors then derive a two-phase approach to compute a policy and approximate z_0 in this augmented problem (section 4).

For that, the 1st phase (and here is where the paper gets a bit confusing) reformulates the augmented problem as a discounted RL where a stochastic policy is used (Algorithm 1). This makes the problem amenable for using PPO and strongly relies on the results and method proposed in Hsu et al. (RSS, 2021 ref [28]). Critically, the value of z_0 is just used as ... another parameter? This is not specified in the paper. It is briefly mentioned in the first reply to Reviewer 186A, but not in clarifying detail:

- "The first phase uses PPO to learn a stochastic policy $\pi$ and corresponding value function $\tilde{V}_{\hat{g}}^\pi$ for different states $x$ and upper-bounds $z_0$."

The 2nd phase (Algorithm 2) serves two purposes: (1) making the stochastic policy obtained during the first phase deterministic, and (2) finally optimizing z_0 (a one dimensional optimization problem for a fixed policy/value function).

The method is then compared against two experimentally in a large variety of tasks showing in all of them competitive or superior performance of the proposed RC-PPO method. The baselines require some design choices that are stated in the paper but more details to guarantee reproducibility would be desirable.

I have several comments:

- A two phase approach could be better motivated/justified. It seems that authors identify RL with the tasks where PPO can be applied to.

- Any mention to z_0 in phase1 is avoided. This is a critical issue. If phase one is supposed to do policy optimization in the augmented problem, what is the value of z_0 chosen then? To me this essentially means solving a problem with the original complexity (finding an accurate optimal cost for the initial state) using RL. Wouldn't the parameter z_0 play a similar role than the threshold used in CMDPs?

- Besides finding z_0^*, the motivation of phase 2 is to make the policy deterministic. Why not just a single phase 1 and decrease the stochasticity of the policy gradually until it becomes deterministic?

- Since the approach strongly relies on ref [28]. Why not compare experimentally with that work?

These are other minor points

- line 192: the augmented dynamic equation is (4), not (5).

- lines 179 and 182, it seems some tildes and hats are missing in the value function.

- line 204: "get 'true'" -> "get the 'true'" and "Thus stationary" -> "Thus, the stationary"

- lines 264: it should be Figure 5 (in the main text), not Figure 6 (in the appendix).

- line 570: "baslines" -> "baselines"

-------------------
The paper was reviewed by three reviewers.

The author's rebuttal period was very useful to clarify several issues and the discussion actually led to additional relevant results that could be used to improve the original submission. Some of the reviewers increase their scores after the clarifications. Finally, the three reviewers agreed that this was an article that should be accepted with different degrees of acceptance by each reviewer (5, 6 and 7).

I have read all the discussions in detail and I am generally satisfied with the authors' responses, including the informative counter examples provided.

I recommend acceptance of the paper. In addition to the improvements developed during the discussion, I suggest my comments above to be addressed in the revised version.